# D-VST: Diffusion Transformer for Pathology-Correct Tone-Controllable Cross-Dye Virtual Staining of Whole Slide Images

**Shurong Yang[1]***   **Dong Wei[2]***   **Yihuang Hu[1]**   **Qiong Peng[1]**
**Hong Liu[1,2]†**   **Yawen Huang[2]**   **Xian Wu[2‡]**   **Yefeng Zheng[2,3]**
**Liansheng Wang[1‡]**

[1]Xiamen University, Xiamen, China    [2]Tencent Jarvis Lab, Shenzhen, China
[3]Westlake University, Hangzhou, China
{xuancheng, huyihuang, qpeng, liuhong}@stu.xmu.edu.cn
{donwei, yawenhuang, kevinxwu, yefengzheng}@tencent.com
lswang@xmu.edu.cn

## Abstract

Diffusion-based virtual staining methods of histopathology images have demonstrated outstanding potential for stain normalization and cross-dye staining (e.g., hematoxylin-eosin to immunohistochemistry). However, achieving pathology-correct cross-dye virtual staining with versatile tone controls poses significant challenges due to the difficulty of decoupling the given pathology and tone conditions. This issue would cause non-pathologic regions to be mistakenly stained like pathologic ones, and vice versa, which we term "pathology leakage." To address this issue, we propose diffusion virtual staining Transformer (D-VST), a new framework with versatile tone control for cross-dye virtual staining. Specifically, we introduce a pathology encoder in conjunction with a tone encoder, combined with a two-stage curriculum learning scheme that decouples pathology and tone conditions, to enable tone control while eliminating pathology leakage. Further, to extend our method for billion-pixel whole slide image (WSI) staining, we introduce a novel frequency-aware adaptive patch sampling strategy for high-quality yet efficient inference of ultra-high resolution images in a zero-shot manner. Integrating these two innovative components facilitates a pathology-correct, tone-controllable, cross-dye WSI virtual staining process. Extensive experiments on three virtual staining tasks that involve translating between four different dyes demonstrate the superiority of our approach in generating high-quality and pathologically accurate images compared to existing methods based on generative adversarial networks and diffusion models. Our code and trained models are available at https://github.com/yangshurong/D-VST.

## 1   Introduction

Histological stainings are used to colorize tissue specimens, making the near-transparent tissue sections visible for pathological observations in clinical diagnostics and research [41]. Different types of dyes manifest different colors in stained tissue and provide complementary information; for example, hematoxylin-eosin (HE) can delineate the cellular structures, whereas immunohistochemistry (IHC)

---

*Co-first authors with equal contribution.

†Work done during an internship at Tencent Jarvis Lab.

‡Corresponding authors.

39th Conference on Neural Information Processing Systems (NeurIPS 2025).

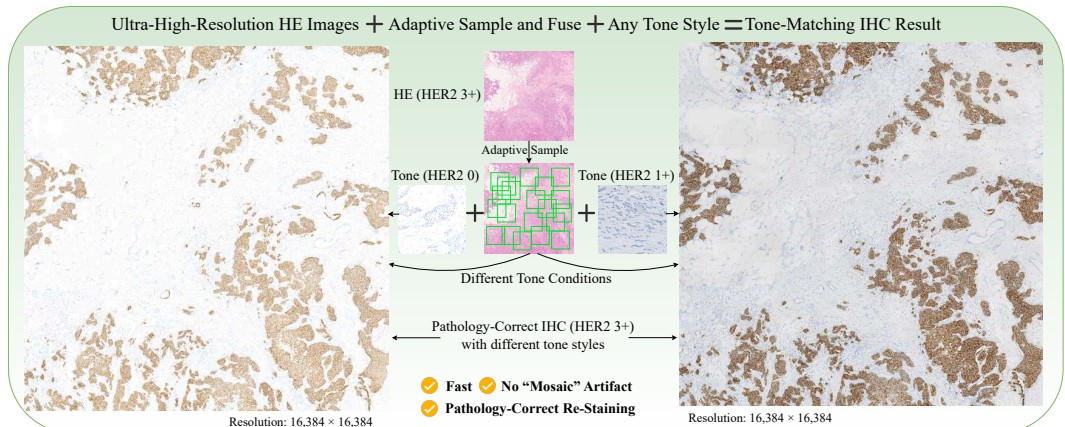

Figure 1: D-VST facilitates efficient (cf. runtime comparison in Table A10), high-quality, tone-controllable, and pathology-correct virtual staining of ultra-high-resolution histopathology images, featuring **adaptive patch sampling** to reduce inference overhead while eliminating mosaic artifacts (cf. Figure 6 and Figure A12); **versatile tone control** by different tone-conditioning images; and **correct pathological status** despite the status of the conditioning images (cf. quantitative and qualitative analysis in Table 3, Figure A10, and Figure A11). HER2 scores: 0: no cancerous lesion, and 1+, 2+, and 3+: increasing severity of cancerous lesions.

renders protein-specific expression to assist in tumor diagnosis and cancer prognosis [2]. However, current chemical protocols allow only one staining per tissue section; additional tissue sections are required for multiple stainings. This adds to the consumption of often limited tissue samples in clinics. In addition, the staining process is time- and chemical-consuming. Therefore, multiple stainings are resource-/labor-intensive and costly [2, 15, 36, 74, 77].

Virtual staining [2] provides a potential solution to multiple stainings—a cost-effective alternative to the conventional chemical process. It digitally "translates" chemically stained histopathology images using computational methods. Researchers leveraged generative adversarial networks (GANs) [37, 100] for virtual staining. Despite notable progress, GANs may encounter significant training challenges, such as mode collapse [79]. Recently, diffusion models have demonstrated superior quality to GANs in controllable image generation [9, 60, 67, 90, 91] and started to be applied to virtual staining of histopathology images. These applications can be divided into two groups: same-dye stain normalization and cross-dye staining. The former addresses the appearance variations in images stained with the same dye [38, 39, 70], likely originating from variations in institute, chemical material, or manual operation [12, 78, 80]. However, the generalization of these methods to the latter—image translation between two different dyes—remains to be investigated.

Cross-dye virtual staining translates histopathology images stained with one dye (the source domain) to new images that look like chemically stained with another (the target domain), e.g., HE to IHC, ideally without structure distortion or pathology status alteration. However, existing methods [20, 31, 33, 40, 53, 54, 86] cannot control the staining tones in the target domain,[4] leading to unpredictable randomness and significant variations in the tones of the virtually stained images. A potential solution is to condition the staining process [67, 95] with desirable tones. However, it is challenging to describe the tones with text. Also, providing the tone condition with an image is more complex than giving structure conditions with Canny edges. Specifically, the tone-conditioning images often contain mixed tone and pathology information. For example, in Figure 2(a), using a cancerous IHC image to condition the virtual staining of a cancer-free HE image may result in erroneous staining that falsely implies the presence of cancer pathology. We term this issue *pathology leakage*. To realize effective tone conditioning without pathology leaks, decoupling the tone and pathology information is crucial (Figure 2(b)).

This work presents diffusion virtual staining Transformer (D-VST), a diffusion model with a Transformer backbone for cross-dye virtual staining of histopathology images (Figure 1). D-VST controls

---

[4]In this work, we refer to the primary color of a type of dye (e.g., the generic pink color of HE staining) as **hue**, and the color variations of that dye as **tones** (e.g., dark to bright pink).

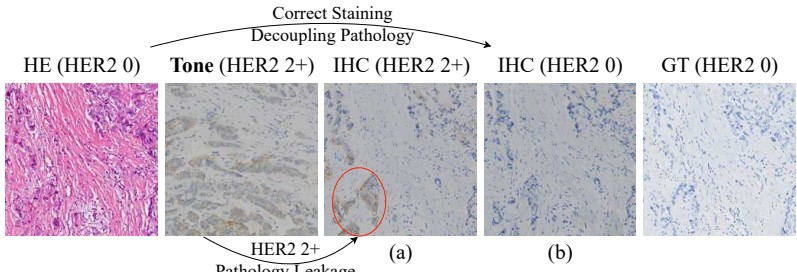

Figure 2: Illustration of **pathology leakage**. (a) Without special treatment, conditioning the virtual staining of a cancer-free HE image with a HER2 2+ IHC image may cause a leak of the cancerous status from the tone-conditioning image to the virtually stained image, leading to pathology-faulty staining. (b) Pathology-correct staining can be achieved by decoupling the tone and pathology conditions. HER2 scores: 0: no cancerous lesion, and 1+, 2+, and 3+: increasing severity of cancerous lesions. GT: ground truth.

the staining tone in the target domain without pathology leaks by adopting separate pathology and tone conditions. Concretely, it relies on the source-domain image to be re-stained for pathological structure conditioning and a target-domain image for tone conditioning. To our knowledge, D-VST represents the first endeavor to realize tone-controllable cross-dye virtual staining of histopathology images with diffusion models. In addition, we design a two-stage curriculum learning scheme to effectively decouple the model's learning of the pathology and tone conditions in a progressive manner. The first stage, pathology extraction, focuses on learning to extract pathology structure information from source-domain images without injecting tone control signals. Then, the second stage, tone injection, adds tone control while diminishing pathology information from the target-domain tone-conditioning image. This involves applying random dropout and Gaussian blur to the tone-conditioning image.

In addition, the virtual staining of large histopathology images like whole slide images (WSIs) requires processing ultra-high-resolution data. However, due to hardware constraints, directly denoising an entire WSI is challenging for diffusion models. Current methods typically divide a WSI into patches, process them individually, and then stitch them together [1, 39, 45, 59, 64]. As the patches are processed independently, this workaround often leads to discrepancies in color, brightness, and contrast between the stitched patches—resulting in the "mosaic" artifact [75]. The mosaic artifact may harm or even invalidate the clinical usability of the virtually stained WSI. To address a similar artifact of content discontinuity in a text-to-panorama application, MultiDiffusion [3] proposed denoising highly overlapping image patches yielded by a sliding-window process separately, followed by fusing the denoising directions by averaging the denoised patches within the overlapped regions. However, unlike natural panoramas, WSIs present significant variations in information density across an image's regions. As a result, the uniform sliding windows in MultiDiffusion may be sub-optimal for WSI virtual staining.

In this work, we present an efficient and high-quality zero-shot inference strategy for virtual staining of WSIs using diffusion models trained under a prevalent resolution, e.g., $512 \times 512$ pixels. Our observation indicates that the mosaic artifact is more pronounced in low-frequency regions of the virtually stained images. Leveraging this insight, we devise a frequency-aware adaptive patch sampling strategy to improve the generation quality of low-frequency regions while controlling computational overhead in high-frequency areas. This strategy enables efficient and rapid virtual staining of billion-pixel WSIs without notable mosaic artifacts, significantly enhancing the capability of our proposed D-VST framework.

**Our contributions** are summarized as follows:

- We propose D-VST, a novel Diffusion Transformer (DiT) [60] based model for histopathology image virtual staining. So far as we know, D-VST is the first diffusion model that realizes tone control for cross-dye virtual staining.

- To address the unwanted pathology leakage issue accompanying the tone control, we design an effective, two-step curriculum learning scheme with separate conditioning branches for pathology and tone.

- We propose an adaptive frequency-aware patch sampling strategy for efficient and high-quality zero-shot staining of billion-pixel WSIs.

- Last but not least, we conduct extensive experiments on three virtual staining tasks involving four dyes to evaluate our D-VST against up-to-date approaches. We also assess a downstream task and perform ablation studies on our method.

## 2    Related work

**GAN-based virtual staining.** Conventional methods predominantly employed GANs [7, 94, 37, 100] for virtual staining of histopathology images. A large amount of work [5, 8, 21, 47, 48, 49, 50, 59, 61, 81, 82, 85, 88] facilitated the transfer of HE to IHC images. [1, 34, 41, 65, 66] showcased generating HE images from formalin fixation and paraffin embedding (FFPE) ones. Moreover, [11, 13, 69] implemented GAN-based image style transfer for stain normalization of histopathology images, effectively mitigating color variations. However, GANs are known to be subject to the mode collapse issue [79] and challenging to train. The emerging diffusion models have recently demonstrated superior training stability, generation controllability, image quality, and versatility to GANs.

**Diffusion-based virtual staining.** Recent advancements in diffusion models [9, 60, 67, 90, 91] have showcased impressive controllable generation capabilities in image synthesis tasks. Various studies [38, 39, 70] employed diffusion models for *stain normalization of histopathology images*. StainDiff [70] proposed self-supervision to facilitate one-to-one color style transfer. StainFuser [39] leveraged the ControlNet [95] to implement fast neural style transfer. [40, 53, 54, 86] examined the potential of diffusion models for *cross-dye histopathology image virtual staining*. [31, 33] implemented cross-dye virtual staining without relying on pathological category labels. VIMs [20] introduced text-controlled protein markers to facilitate virtual staining across multiple pathological categories. However, these methods cannot control the tones for virtual staining, resulting in unpredictable appearance variations among the re-stained images. In contrast, our method controls the staining tone with a target-domain tone-conditioning image.

**WSI generation.** Virtual staining of WSIs, characterized by ultra-high resolution, presents significant challenges to diffusion models. [26, 30, 35, 52, 71, 72, 84, 87, 89, 96] introduced additional global control signals for direct high-resolution image generation by diffusion models. Yet, this approach does not apply to WSIs due to computational limitations. Alternatively, the sliding window strategy [3, 17, 19, 24, 28, 46, 75] offers a viable means. However, this strategy is impeded by substantial inference times due to the highly redundant sliding windows with small sliding steps, or subject to a performance drop in image quality with large sliding steps. [22, 99] proposed selecting patches at varying timesteps to mitigate long inference duration, yet this mechanism may introduce instability to the generated outcome. To facilitate efficient and high-quality WSI virtual staining, we propose a novel adaptive patch sampling strategy based on image frequency variations.

## 3    Method

The framework of our method is shown in Figure 3 (left). Primarily, D-VST tailors and extends the Diffusion Transformer (DiT) [60] in PixArt-$\alpha$ [9] as its denoising Transformer. A pathology encoder encodes the source image into pathological structure embeddings, which are injected into the denoising Transformer as the structure condition after concatenating with the noise latent. Meanwhile, a tone encoder (which we use the pretrained Vision Transformer (ViT) [18] in CLIP [63]) encodes an auxiliary, tone-conditioning image from the target domain into tone embeddings. The tone embeddings are also injected into the denoising Transformer via multi-head tone attention. The denoising Transformer integrates the structure and tone conditions and denoises toward a re-stained target histopathology image (latent).

The pipeline of D-VST proposes innovative designs for both training and inference schemes. To prevent pathology leakage from the tone-conditioning image, we decompose the intricate virtual staining task into a curriculum learning [4] streamline for progressive training [68]. To eliminate the mosaic artifact at a low computational cost while staining high-resolution images, we propose a frequency-aware adaptive patch sampling strategy for efficient inference.

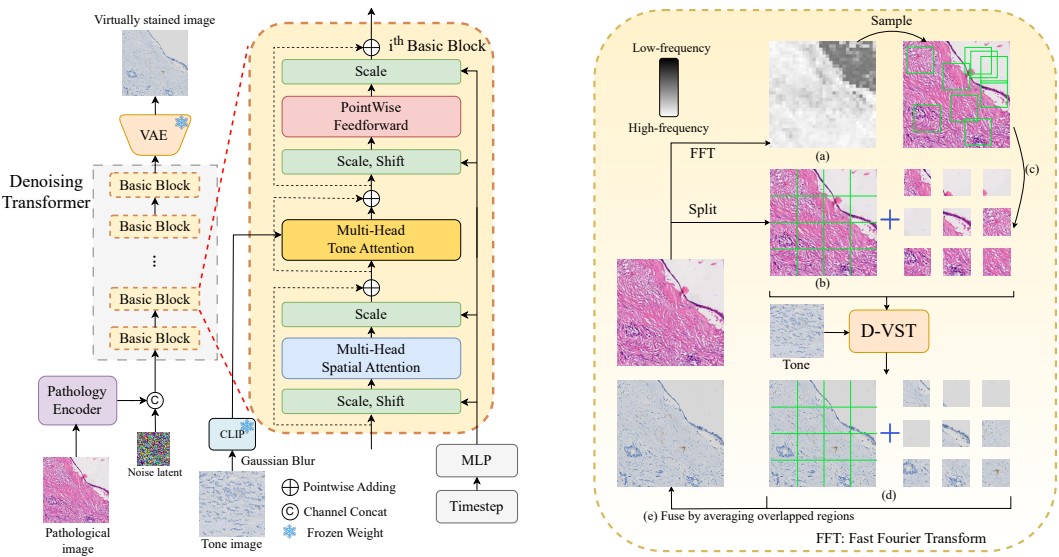

Figure 3: Left: **framework of the proposed D-VST**. Right: **frequency-aware adaptive patch sampling.** (a) FFT-based local frequency computing. (b) Covering the entire image with tiled, non-overlapping patches. (c) Additional patches sampled according to local frequency. We show only nine patches here to illustrate that patches are more likely to be sampled from low- than high-frequency regions. In fact, all sampled patches cover seven times the area of the input image. (d) Individually denoising each patch. (e) Fusing by averaging overlapped regions.

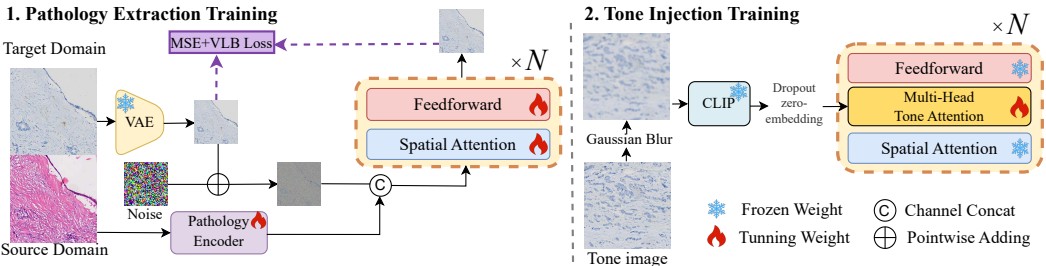

Figure 4: The **two-stage curriculum learning scheme** decouples pathology and tone conditions. Initially, only the pathology condition is input to the model, emphasizing accurate pathology extraction while ignoring the significant hue difference between the source and target domains. Next, we add tone condition via cross-attention, with random dropout and Gaussian blur to diminish pathology leakage from the tone-conditioning image.

## 3.1 Curriculum learning scheme decoupling pathology and tone conditions

As introduced in Figure 2, training with mingled pathology and tone controls often causes pathological status to leak through tone conditioning, leading to incorrect staining outcomes. This occurs because the model confuses the purposes of the conditioning images and mistakes the tone-conditioning image for the source of the pathology status. Thus, the key to effectively preventing pathology leakage is to decouple pathology and tone conditions by making the model learn both control signals precisely. For this purpose, we design a novel two-stage curriculum learning [4] scheme. In the first stage, pathology conditioning is first learned alone. Then, tone conditioning is introduced in the second stage for further joint training.

**Training stage 1: pathology extraction.** Figure 4 (left) illustrates the first training stage. In this stage, we only feed the pathology condition into the model via channel-wise concatenation with the noise latent. Meanwhile, the denoising Transformer includes only the spatial attention and feedforward modules. The training goal is to denoise the corrupted target-domain image (obtained by adding Gaussian noise as in typical diffusion processes) in the latent space of a pretrained variational

autoencoder (VAE)[5] [43], guided by the pathology structure embedding extracted from the *paired* pathology-conditioning source-domain image. We follow [60] to train the model with the hybrid mean squared error (MSE) and variational lower bound (VLB) losses [55], where the former learns to predict the sampled noise and the latter learns variances of the reverse diffusion process (cf. Appendix for more details). Thus, the model learns to effectively extract and utilize the complex pathology information in the pathology-conditioning image in this stage, ignoring the significant hue difference between the source and target domains.

The pathology encoder uses a lightweight convolutional network with $4 \times 4$ kernels, $2 \times 2$ strides, and four layers of 16, 32, 64, and 128 channels. It encodes the pathology-conditioning image into an embedding of the same shape as the noise latent. The embedding is concatenated with the noise latent along the channel dimension and input to the denoising Transformer. Compared to the ControlNet [95] architecture used in StainFuser [39], our lightweight pathology encoder is equally effective in capturing detailed pathological information while substantially reducing model complexity and computational cost.

**Training stage 2: tone injection.** The second stage introduces tone conditioning into the model (Figure 4 (right)). A random patch from the same WSI but *different from (thus not paired with)* the target-domain image is used for tone conditioning. It provides a precise tone style but not necessarily pathology information of the target image (different patches of a WSI may present distinct pathological statuses). To emphasize tone features while minimizing pathological structural information, we first apply a Gaussian blur to the tone-conditioning image. Then, we utilize the pretrained ViT [18] in OpenAI-CLIP [63] to encode the blurred image into a tone embedding. The OpenAI-CLIP ViT was trained on massive data of versatile *colors*, making it a proper *tone* encoder for *optical* pathology images (cf. Appendix for a comparison to the PathCLIP [76]). Next, in the middle of the frozen feedforward and spatial attention modules, we insert a multi-head tone attention module into each unit block of the denoising Transformer trained in the first stage. Lastly, using cross-attention, we inject the tone embedding into the denoising Transformer via the inserted tone attention modules. To further reduce pathology leakage, we apply a random dropout to the tone embedding by replacing it with a zero embedding. The dropout rate is set to 20% according to preliminary trials.[6] The same training losses as in the first stage are used. Ablation studies confirm the efficacy of the tone encoder's components (Table 4).

## 3.2 Frequency-aware adaptive patch sampling

From the "No-overlap" column of Figure 6, we observe that the mosaic artifact is more pronounced in low-frequency regions of the virtually stained images. This occurs because, unlike areas with complex textures that contain abundant clues guiding the virtual staining process, low-frequency regions require more overlapping patches for consistent denoising results. To improve the generation quality of the low-frequency regions while simultaneously controlling computational overhead, we propose a frequency-aware adaptive patch sampling strategy. As the premise, we divide the input image into a grid of $n \times n$ squares. We set $n = 32$ through a grid search (cf. Appendix). Then, for each square $I$, we compute the natural logarithm of the magnitudes of its fast Fourier transform (FFT; Figure 3 (right)-(a)):

$$L = \log \left[ \operatorname{abs} \left( \operatorname{FFT}(I) \right) \right]. \tag{1}$$

Next, we calculate the pixel-wise mean of $L$, denoted by $l$, as the frequency statistic for the square. Finally, we convert $l$ to sampling probability by:

$$p_i = \frac{(l^{\max} + l^{\min} - l_i)^\alpha}{\sum_{i=1}^{n^2} (l^{\max} + l^{\min} - l_i)^\alpha}, \tag{2}$$

where $l^{\max}$ and $l^{\min}$ are the maximum and minimum $l$ values of all $n^2$ squares, $(l^{\max} + l^{\min} - l_i)$ makes low-frequency squares more likely to be sampled, and $\alpha \in \mathbb{Z}^+$ is a hyperparameter controlling the difference in sampling probabilities between low- and high-frequency squares.

---

[5]In our preliminary experiments, when applying the VAE of Stable Diffusion (SD) 1.5 [67] to pathology image reconstruction, we obtained an average peak signal-to-noise ratio of $\sim$28 dB, comparable to the performance on natural images ($\sim$25 dB). Therefore, we use the SD 1.5 VAE in this work.

[6]Further optimization via rigorous ablation study may lead to better performance; we leave it for future work.

To ensure the entire input image is stained, we first fully cover the image with tiled, non-overlapping patches (Figure 3 (right)-(b); note the patches are larger than the squares for FFT). Next, we sample additional patches according to the sampling probabilities in Equation (2) (Figure 3 (right)-(c)): a square $I$ is first selected according to $p_i$, and a random pixel within $I$ is subsequently chosen as the center of an additional patch. This way, more patches are adaptively sampled where necessary, whereas fewer patches are sampled from high-frequency regions to maintain reasonable computational costs. In this work, we define an integer $\beta$ as the ratio of the total area of tiled and sampled patches to the area of the stained image. The larger $\beta$ is, the more patches are sampled (the special case of no-overlap sliding windows has $\beta = 1$). The patches are processed separately in every denoising step, and the denoised patches are fused in overlapping areas by averaging (Figure 3 (right)-(d) and (e)) [3]. Experiments show that compared with MultiDiffusion [3], our strategy improves the virtual staining quality with only 12.5% computational overhead ($\beta = 8$ versus $64$; cf. Table 2).

## 4    Experiments

**Datasets and evaluation metrics.** We evaluate D-VST on three datasets to comprehensively validate its performance on the virtual staining of various dyes. RegH2I [61] comprises 2,592 pairs of registered images for HE to IHC staining (HE2IHC), [34] includes 5,098 pairs of aligned images for FFPE to HE (FFPE2HE) staining, and HEMIT [5] contains 5,292 matched image pairs for HE to multiplex immunohistochemistry (mIHC) staining (HE2mIHC). These datasets include two organs/cancer types: breast cancer (HE2IHC and FFPE2HE) and colon cancer (HE2mIHC). We use all datasets' official train/test/validation splits. Unless otherwise specified, we report performance on cropped images of $1024 \times 1024$ pixels as in [5, 34, 61], and use a random patch from the same WSI but *not overlapping* with the target image for tone conditioning. Note that the tone-conditioning and target images may present different pathological statuses, and the no-overlap requirement ensures no structural leak.

Following previous works [5, 34, 61], we employ the structural similarity index measure (SSIM) and peak signal-to-noise ratio (PSNR) for quality assessment of the virtually stained images. However, for paired histopathology images stained with different dyes, perfect pixel-to-pixel matching is practically impossible even after registration (see Appendix for more explanation). Therefore, for an appropriate evaluation, we additionally employ three metrics that are more perceptually relevant than the conventional SSIM and PSNR: deep image structure and texture similarity (DISTS) [16], Fréchet inception distance (FID), and kernel inception distance (KID) [6].

**Implementation details.** All experiments are conducted in Python 3.10.0 with PyTorch 2.0.0 [58] on a GPU with 80 GB of memory. We follow [9] to use DiT-XL/2 [60] as the base network architecture for our denoising Transformer, and the pretrained parameters from [9]. We employ the AdamW [44] optimizer with a learning rate of $10^{-5}$ and a batch size 32. We train for 30,000 steps for the pathology extraction stage, and an additional 10,000 steps for the tone injection stage. The diffusion time $T$ is set to 1000. Our model trains and infers at the resolution of $512 \times 512$ pixels. For virtual staining of larger images, we use the model to denoise $512 \times 512$ patches sampled according to the proposed frequency-aware adaptive patch sampling strategy (cf. Section 3.2), and fuse the patch-wise outcomes by averaging overlapped regions [3]. Unless otherwise specified, we set $\alpha = 1$ and $\beta = 8$ for the adaptive sampling. Our code and trained models are available at https://github.com/yangshurong/D-VST.

**Comparison with state-of-the-art (SOTA).** We compare our D-VST with classical GAN-based image translation methods: CycleGAN [100], pix2pix [37] and pix2pixHD [83]; medical image diffusion model: SynDiff [56]; and SOTA GAN/diffusion models specialized in histopathology image virtual staining: [34] and [61]/StainFuser [39]. The comparisons with [34] and [61] are exclusively on the FFPE2HE [34] and HE2IHC [61] datasets, respectively, since the two methods were designed for the specific tasks. As shown in Table 1, D-VST achieves the best performance for all metrics on the HE2IHC and HE2mIHC datasets, indicating that the histopathology images virtually stained by D-VST are superior both perceptually and structurally. On the FFPE2HE dataset, D-VST yields slightly inferior PSNR and SSIM to the best numbers, yet is still competitive. We conjecture this is because the micro-level correspondence (pixel- and structure-wise) between the paired images in this dataset is not as good as the other two. Notwithstanding, D-VST again achieves the best performance

Table 1: Evaluation of various methods on three different datasets and virtual staining tasks.

| Method | DISTS↓ | FID↓ | KID↓ | PSNR↑ | SSIM↑ |
|---|---|---|---|---|---|
| *HE2IHC* [61] | | | | | |
| Pix2pix [37] | 0.192 | 47.77 | 0.0237 | 18.04±3.693 | 0.401±0.126 |
| Pix2pixHD [83] | 0.191 | 41.77 | 0.0123 | 18.06±3.733 | 0.386±0.128 |
| CycleGAN [100] | 0.212 | 40.91 | 0.0062 | 17.01±3.524 | 0.365±0.119 |
| [61] | 0.174 | 33.92 | 0.0058 | 18.02±3.706 | 0.385±0.125 |
| SynDiff [56] | 0.348 | 225.3 | 0.2282 | 18.09±4.049 | 0.404±0.114 |
| StainFuser [39] | 0.255 | 104.5 | 0.0791 | 17.30±3.949 | 0.401±0.148 |
| D-VST | **0.154** | **33.16** | **0.0055** | **18.11±3.874** | **0.407±0.136** |
| *HE2mIHC* [5] | | | | | |
| Pix2pix[37] | 0.133 | 29.95 | 0.0058 | 27.26±3.903 | 0.855±0.063 |
| Pix2pixHD [83] | 0.170 | 28.92 | 0.0086 | 27.65±3.916 | 0.816±0.062 |
| CycleGAN [100] | 0.300 | 83.16 | 0.0365 | 20.15±1.663 | 0.520±0.049 |
| SynDiff [56] | 0.318 | 316.1 | 0.4057 | 20.05±1.423 | 0.709±0.038 |
| StainFuser [39] | 0.289 | 99.20 | 0.0690 | 20.27±2.087 | 0.309±0.040 |
| D-VST | **0.106** | **20.36** | **0.0016** | **28.01±4.123** | **0.861±0.045** |
| *FFPE2HE* [34] | | | | | |
| Pix2pix [37] | 0.109 | 17.88 | 0.0008 | 18.34±2.009 | 0.536±0.113 |
| Pix2pixHD [83] | 0.091 | 15.26 | 0.0008 | 19.08±2.046 | 0.586±0.106 |
| CycleGAN [100] | 0.171 | 35.36 | 0.0087 | 14.47±2.045 | 0.363±0.152 |
| [34] | 0.126 | 31.56 | 0.0101 | **19.86±2.082** | **0.644±0.098** |
| SynDiff [56] | 0.228 | 69.44 | 0.0329 | 12.15±1.787 | 0.336±0.146 |
| StainFuser [39] | 0.200 | 64.98 | 0.0301 | 12.46±1.851 | 0.263±0.157 |
| D-VST | **0.090** | **14.26** | **0.0005** | 17.98±2.165 | 0.538±0.117 |

Table 2: Evaluation of sampling strategies for zero-shot staining of images larger than the training resolution of diffusion models. $\beta$ is the ratio of the total patch area to the area of the virtually stained image. Given a fixed patch size, the larger $\beta$ is, the more patches are sampled, thus the higher computational cost.

| Sample strategy | $\beta$ | DISTS↓ | FID↓ | KID↓ | PSNR↑ | SSIM↑ |
|---|---|---|---|---|---|---|
| *HE2IHC* [61] | | | | | | |
| No-overlap | 1 | 0.1594 | 37.748 | 0.0092 | 18.01±3.868 | 0.406±0.136 |
| SpotDiffusion [22] | 1 | 0.2256 | 93.349 | 0.0664 | 14.08±2.815 | 0.297±0.117 |
| MultiDiffusion [3] | 64 | 0.1566 | 34.661 | 0.0061 | 18.09±3.877 | 0.406±0.136 |
| D-VST | 8 | **0.1548** | **33.162** | **0.0055** | **18.11±3.874** | **0.407±0.136** |
| *HE2mIHC* [5] | | | | | | |
| No-overlap | 1 | 0.1063 | 21.866 | 0.0021 | 27.72±4.534 | 0.852±0.048 |
| SpotDiffusion [22] | 1 | 0.4285 | 111.35 | 0.0937 | 10.87±3.717 | 0.373±0.059 |
| MultiDiffusion [3] | 64 | 0.1069 | 20.365 | 0.0016 | 27.96±4.260 | 0.858 ±0.045 |
| D-VST | 8 | **0.1062** | **20.361** | **0.0016** | **28.01±4.123** | **0.861±0.045** |
| *FFPE2HE* [34] | | | | | | |
| No-overlap | 1 | 0.0929 | 16.403 | 0.0012 | 17.60±2.183 | 0.517±0.120 |
| SpotDiffusion [22] | 1 | 0.1483 | 48.281 | 0.0256 | 15.41±1.408 | 0.467±0.109 |
| MultiDiffusion [3] | 64 | 0.0910 | 14.960 | 0.0007 | 17.87±2.165 | 0.531±0.118 |
| D-VST | 8 | **0.0900** | **14.263** | **0.0005** | **17.98±2.166** | **0.538±0.117** |

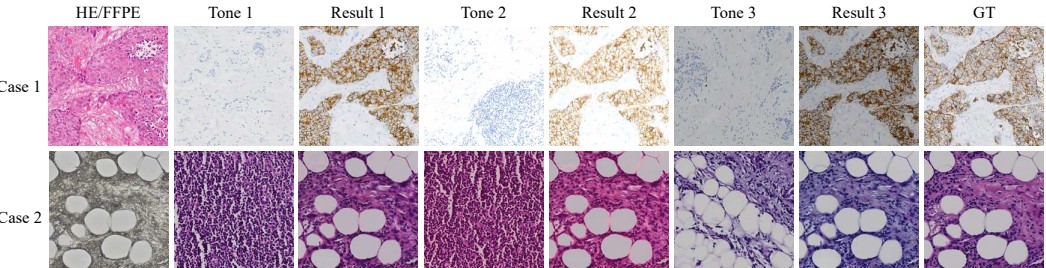

Figure 5: Qualitative results for **tone control**. Case 1 is from the HE2IHC [61] dataset, whereas Case 2 is from the FFPE2HE [34] dataset. Despite substantial discrepancies in pathological status between the pathology- and tone-conditioning images, the pathological status is correctly transferred from the pathology-conditioning images to the virtually stained ones. For example, in Case 1, the HER2 score of the HE image is 3+, while the scores of Tone 1, 2, and 3 are all 0. GT: ground truth.

for the three perception-oriented metrics (DISTS, FID, and KID). In general, D-VST demonstrates strong capabilities in virtually staining high-quality histopathology images of versatile dyes.

**Tone control and downstream task.** As the HE2mIHC [5] dataset has undergone color normalization and thus cannot provide varying tone conditions, we conduct qualitative tone control experiments on the other two datasets. As shown in Figure 5, when conditioned on histopathology images of various tones of another dye, the virtually stained images exhibit varying tones matching the tone-conditioning images while maintaining the same pathological status as the source pathology-conditioning images. For example, Case 1 illustrates that even when there are substantial discrepancies in pathological status between the pathology- and tone-conditioning images, the pathological status is still correctly transferred from the pathology-conditioning image to the virtually stained ones. These observations indicate that our D-VST can effectively prevent pathology leakage for tone-conditioned cross-dye virtual staining. We provide more visualizations and comparisons with other methods in Appendix.

To further *quantitatively* validate our method's effectiveness in pathology leakage prevention while using image-based tone conditioning, we perform a downstream classification task on HE2IHC [61]. Concretely, we further split the 600 official test pairs into a sub-train and sub-test set of 480 and 120 pairs, respectively. Then, we train a ResNet50 [29] classifier on the IHC images of the sub-train set, with labels corresponding to the four HER2 scores (HER2 0: no cancerous lesions, and 1+, 2+, and 3+: increasing severity of cancerous lesions, with higher scores indicating more pronounced lesions and more advanced disease stages). Next, for each HE image in the sub-test set, we randomly select an IHC image in the sub-test set that is not paired with the specific HE image as the tone condition for virtual staining. Lastly, we apply the trained classifier to the virtually stained IHC

Table 3: Evaluation of downstream classification task on the HE2IHC [61] dataset.

| Method | ACC↑ | F1↑ | Precision↑ | Recall↑ |
|---|---|---|---|---|
| Pix2pix [37] | 0.8750 | 0.8755 | 0.8916 | 0.8779 |
| Pix2pixHD [83] | 0.8500 | 0.8508 | 0.8655 | 0.8384 |
| CycleGAN [100] | 0.5583 | 0.5609 | 0.5600 | 0.5512 |
| [61] | 0.8917 | 0.8925 | 0.8930 | 0.8847 |
| SynDiff [56] | 0.2167 | 0.0884 | 0.3025 | 0.2589 |
| StainFuser [39] | 0.7250 | 0.7324 | 0.7331 | 0.7230 |
| D-VST | **0.9417** | **0.9430** | **0.9470** | **0.9388** |
| Real IHC images | 0.9500 | 0.9506 | 0.9530 | 0.9488 |

Table 4: Ablation study on the HE2IHC [61] dataset with both image generation and downstream task metrics.

| Metric | w/o Dropout | w/o Gaussian | w/o Curriculum | D-VST |
|---|---|---|---|---|
| ACC↑ | 0.9250 | 0.7833 | 0.7667 | **0.9417** |
| F1↑ | 0.9265 | 0.7710 | 0.7672 | **0.9430** |
| Precision↑ | 0.9292 | 0.8425 | 0.7720 | **0.9470** |
| Recall↑ | 0.9250 | 0.7585 | 0.7645 | **0.9388** |
| DISTS↓ | 0.1589 | 0.2372 | 0.1729 | **0.1548** |
| FID↓ | 35.815 | 55.148 | 38.957 | **33.162** |
| KID↓ | 0.0070 | 0.0290 | 0.0110 | **0.0055** |
| PSNR↑ | 17.451±3.629 | 15.852±3.440 | 17.587±3.574 | **18.113±3.874** |
| SSIM↑ | 0.3904±0.142 | **0.4361±0.145** | 0.4012±0.137 | 0.4074±0.136 |

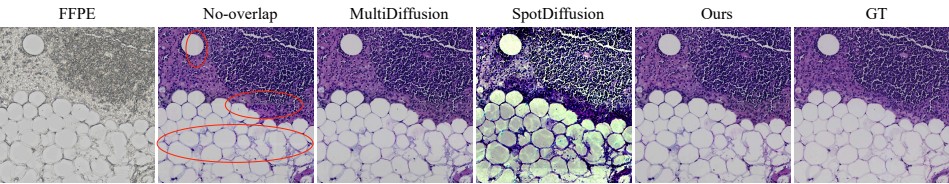

Figure 6: Comparison of sampling strategies for zero-shot virtual staining of large histopathology images. Red ellipses outline regions where the mosaic artifacts are prominent (best viewed zoomed in). GT: HE ground truth. More is provided in Appendix.

images. Intuitively, if the pathology leakage happens, the classification results would notably deviate from directly classifying the real IHC images in the sub-test set. Table 3 shows that our classifier achieves high performance on real IHC images, and more importantly, the performance on virtually stained IHC images by D-VST is also high and closely matches that on real ones. Meanwhile, D-VST obtains substantially better results than the compared methods. These results demonstrate that our method successfully decouples pathology and tone information for tone-conditioned cross-dye virtual staining, and its virtually stained images are of high quality for potential clinical use.

**Sampling strategy for zero-shot virtual staining of large histopathology images.** As described in the implementation details, our model trains and infers at the resolution of 512×512 pixels. For the virtual staining of larger images, we use the model to denoise 512×512 patch samples, followed by patch fusion. Here, we compare several sampling strategies in terms of performance and computational cost: no-overlap (sliding windows without overlap), SpotDiffusion [22] (sampling sliding windows that vary with timesteps), MultiDiffusion [3] (sliding windows with a high overlap ratio), and our proposed frequency-aware adaptive patch sampling. Since different sampling strategies are evaluated on the same network (our proposed), which takes about 1.081s and 6.4 GB memory to infer a patch of 512×512 pixels on our hardware, their relative computational costs can be compared by $\beta$ values. We conduct quantitative evaluations on the HE2IHC, FFPE2HE, and HE2mIHC datasets, generating images of 1024×1024 pixels. As shown in Table 2, although no-overlap and SpotDiffusion incur the least computational cost, their performance is the worst. MultiDiffusion improves all evaluated metrics, though at 64 times the inference cost. In contrast, our strategy achieves the best performance for all metrics on the three datasets, while incurring only 1/8 of the inference computation of MultiDiffusion. For more insights, we additionally evaluate MultiDiffusion with the same $\beta = 8$ as ours on HE2IHC. Its FID degrades from 34.66 to 34.99, markedly inferior to our 33.16. These results demonstrate that our frequency-aware adaptive sampling strategy is not only highly efficient but also capable of boosting the quality of virtual staining.

Figure 6 shows example results by the compared methods for qualitative analysis, virtually stained at the resolution of 2048×2048 pixels. No-overlap exhibits noticeable mosaic artifacts, whereas SpotDiffusion presents anomalous tones—accounting for their unsatisfactory quantitative results. With only 1/8 of the computational cost of MultiDiffusion, our strategy produces images of equal visual quality to MultiDiffusion. We have also applied D-VST to the virtual staining of WSIs of 0.2–1.3 billion pixels (16,000×15,000 to 40,000×32,000 pixels). However, as far as we know, no suitable WSI dataset is currently available for reliable quantitative evaluation at scale. Therefore, we only show the qualitative results in Appendix for an observational study.

**Ablation studies.** In Section 3.1, we have proposed two-stage curriculum learning, Gaussian blur of tone-conditioning images, and random dropout of the tone condition to realize pathology and

tone decoupling and prevent pathology leakage. Here, we ablate one of them at a time (denoted by w/o Curriculum, w/o Gaussian, and w/o Dropout) to study their efficacy on the HE2IHC dataset using both image generation and downstream classification metrics. Table 4 shows that removing any of them leads to overall declines in all metrics (the only exception is the SSIM w/o Gaussian). Especially, removing curriculum learning results in the most significant performance drops in three of the four classification metrics. These results suggest that these components, especially the two-stage curriculum learning scheme, effectively boost the performance of histopathology image virtual staining. This is achieved by improving the perceptual quality via effective pathology and tone condition disentanglement, thus fulfilling our design.

The Appendix includes further experiments determining the values of the hyperparameters $\alpha$ and $\beta$ in the frequency-aware adaptive patch sampling (cf. Section 3.2).

## 5  Conclusion

This work presented D-VST, a diffusion Transformer based framework for efficient, high-quality, and pathology-preserving cross-dye virtual staining of histopathology images with up to more than a billion pixels. Extensive experiments on three virtual staining tasks involving four types of dyes and a downstream cancer status classification task validated D-VST's promising performance. Facilitating efficient tone-controllable virtual staining, D-VST has the potential to make a broad impact on algorithm development and the clinical pipeline of histopathology image analysis.

**Limitations and future work.** Our D-VST facilitates efficient virtual staining of ultra-high-resolution histopathology images like WSIs by the proposed frequency-aware adaptive patch sampling strategy. However, its inference speed is still constrained by the multi-step denoising process inherent in diffusion models [32]. Inspired by [42, 51, 73], we plan to optimize the denoising scheduler and reduce inference steps by flow rectification and consistency models, and further reduce computation overhead and accelerate inference by model pruning and distillation retraining [93].

Obtaining paired cross-stain training data can be challenging and costly in real-world workflows, which may limit the scalability and applicability of D-VST. While such data can improve virtual staining performance with paired correspondence, unpaired data is substantially more scalable due to orders of magnitude larger amounts. In future work, we plan to explore benefiting from both the scalability of unpaired data and the quality of paired data via a combination of D-VST and approaches [92] like CycleGAN-Turbo [27, 57], which enable diffusion models to learn from unpaired data.

In this work, we have attempted to fine-tune the VAE alongside the Diffusion Transformer in our experiments but obtained mixed results (similar PSNR and SSIM with poorer DISTS, FID, and KID), likely due to the limited training data. In the future, we plan to explore whether substituting the VAE with a histopathology-image-pretrained counterpart would further enhance our framework's performance.

Lastly, future work will investigate D-VST's benefits for downstream segmentation tasks [97].

## Acknowledgments

This work was supported by the National Natural Science Foundation of China (Grant No. 62371409).

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

## A1   Training losses

We briefly describe the losses for training our framework. In denoising diffusion probabilistic models (DDPMs) [32], a forward process gradually applies noise to real data $x_0$: $q(x_t|x_0) = \mathcal{N}(x_t; \sqrt{\bar{\alpha}_t}x_0, (1 - \bar{\alpha}_t)\mathbf{I})$, where $t$ is the time step and constants $\bar{\alpha}_t$ are hyper-parameters. Applying the reparameterization trick, $x_t$ can be sampled by $x_t = \sqrt{\bar{\alpha}_t}x_0 + \sqrt{1 - \bar{\alpha}_t}\epsilon_t$, where $\epsilon_t \sim \mathcal{N}(0, \mathbf{I})$. Inversely, a model learns the reverse process to gradually restore the noise-corrupted real data by $p_\theta(x_{t-1}|x_t) = \mathcal{N}(\mu_\theta(x_t), \Sigma_\theta(x_t))$, using neural networks to predict the statistics of $p_\theta$. The model is trained with the variational lower bound (VLB) of the log-likelihood of $x_0$, which can be written as (omitting a training-irrelevant term): $\mathcal{L}_{\text{VLB}} = -\log p_\theta(x_0|x_1) + \sum_t \mathcal{D}_{\text{KL}}\big(q(x_{t-1}|x_t, x_0)\|p_\theta(x_{t-1}|x_t)\big)$, where $\mathcal{D}_{\text{KL}}$ is the Kullback-Leibler (KL) divergence loss. By parameterizing $\mu_\theta$ as a noise prediction network $\epsilon_\theta$, the model can be trained with the mean-squared error loss between the predicted noise $\epsilon_\theta(x_t)$ and the ground-truth sampled Gaussian noise $\epsilon_t$: $\mathcal{L}_{\text{MSE}} = \|\epsilon_\theta(x_t) - \epsilon_t\|_2^2$. Meanwhile, to learn the covariance $\Sigma_\theta$, the full $\mathcal{L}_{\text{VLB}}$ needs to be optimized. We follow [55] to train $\epsilon_\theta$ with $\mathcal{L}_{\text{MSE}}$, and $\Sigma_\theta$ with $\mathcal{L}_{\text{VLB}}$. Since the training losses are not a focus of this paper, we refer interested readers to [55] for more details.

## A2   Justification for non-perfect paired data

In virtual staining tasks, paired images are typically obtained from consecutive tissue sections and algorithmically registered. Although the alignment does not perfectly match all pixels, most are closely aligned and thus valid for structural correspondence learning. Before ours, many methods successfully trained their virtual staining models on the datasets used in this work [5, 34, 61].

In the main text, we conjectured that D-VST's lower PSNR/SSIM versus [34] on FFPE2HE "is because the micro-level correspondence (pixel- and structure-wise) between the paired images in this dataset is not as good as the other two." Although this misalignment is inherent in all datasets used in this work due to the consecutive slicing and chemical staining process, we visually find it more serious in the FFPE2HE dataset. To quantify the structural (mis)alignment between paired images, we resort to the following procedures. We apply the Canny edge detector and compute the Hausdorff distance, intersection-over-union (IoU), and Dice similarity between edge maps of the source and target images. Intuitively, lower Hausdorff distance and higher IoU and Dice metrics indicate better structural alignment. As shown in Table A5, FFPE2HE [34] shows consistently worse alignment metrics, supporting our conjecture. In particular, the Hausdorff distance measures the maximum deviation between two point sets, highlighting the worst-case alignment errors. Thus, the substantially larger Hausdorff distances indicate more extreme misalignments.

Table A5: Quantification of structural (mis)alignment between paired images in the HE2IHC and FFPE2HE datasets using Hausdorff distance, intersection-over-union (IoU), and Dice similarity.

| Datasets | Hausdorff↓ | IoU↑ | Dice↑ |
|---|---|---|---|
| HE2IHC | $30.92\pm_{12.05}$ | $0.150\pm_{0.026}$ | $0.260\pm_{0.041}$ |
| FFPE2HE | $36.63\pm_{33.13}$ | $0.138\pm_{0.011}$ | $0.242\pm_{0.016}$ |

## A3   Additional experiments

**Choice of tone encoder.** For the tone encoder, we experimented with three image encoders: one pretrained in OpenAI-CLIP [63] on massive data of broad spectrums; one pretrained in PathCLIP [76] on 207K high-quality pathology image–caption pairs; and one pretrained in UNI [10] on over 200 million pathology HE and IHC images. Table A6 presents their performance on the HE2IHC dataset [61]. UNI and OpenAI-CLIP demonstrate comparable performance, and clearly outperform PathCLIP in perception-oriented metrics while remaining comparable in PSNR and SSIM. We conjecture that the difference in performance may be partly attributed to the function of the tone encoder. On the one hand, while PathCLIP may be better prepared for downstream tasks on pathology images (e.g., classification), the tone encoder focuses more on color perception. As a result, OpenAI-CLIP may be more suitable due to its more significant amount of training data that inherently includes more

versatile color variations. Thus, we use OpenAI-CLIP within our D-VST framework. On the other hand, while UNI offers strong pathological feature extraction, OpenAI-CLIP suffices for capturing tone information in D-VST. Hence, pathology foundation models trained on multi-stains like UNI are also excellent choices for the proposed D-VST framework. However, it is important to note that even when using PathCLIP as the tone encoder, our performance remains competitive with other methods in Table 1 of the main text.

Table A6: Performance comparison of using PathCLIP [76], UNI [10] and OpenAI-CLIP [63] as the tone encoder on the HE2IHC [61] dataset.

| Tone encoder | DISTS↓ | FID↓ | KID↓ | PSNR↑ | SSIM↑ |
|---|---|---|---|---|---|
| PathCLIP | 0.190 | 40.01 | 0.0086 | 17.49±3.717 | **0.411**±**0.136** |
| UNI | 0.157 | **33.13** | **0.0048** | 18.03±3.842 | 0.404±0.130 |
| OpenAI-CLIP | **0.154** | 33.16 | 0.0055 | **18.11**±**3.874** | 0.407±0.136 |

**Influence of hyper-parameter $n$ for square split.** The input images to be re-stained are divided into a grid of $n \times n$ squares for local frequency estimate. We vary $n$ from 4 to 128 and show the FIDs for HE2IHC [61] in Table A7. When $n \in \{16, 32, 64\}$, the results are the best and stable, whereas $n$ being too small or large deteriorates the performance. The empirical guideline is to select $n$ properly so that each square contains enough pixels for a reliable frequency estimate but not too many pixels to remain a local estimate. We use $n = 32$ in our paper.

Table A7: Performance in FID with varying values for the hyper-parameter $n$ on the HE2IHC [61] dataset.

| $n$ | 4 | 8 | 16 | 32 | 64 | 128 |
|---|---|---|---|---|---|---|
| FID↓ | 33.452 | 33.444 | 33.267 | **33.162** | 33.323 | 33.548 |

**Ablation study on $\alpha$ and $\beta$.** $\alpha$ and $\beta$ are important hyper-parameters in our proposed frequency-aware adaptive patch sampling. Concretely, $\alpha$ controls the sampling probability discrepancy between low- and high-frequency regions, whereas $\beta$ trades off between computational cost and image quality. Figure A7 presents the impact of varying $\alpha$ and $\beta$ values on the model performance focused on FID, whose variations are more evident among the perception-oriented metrics.

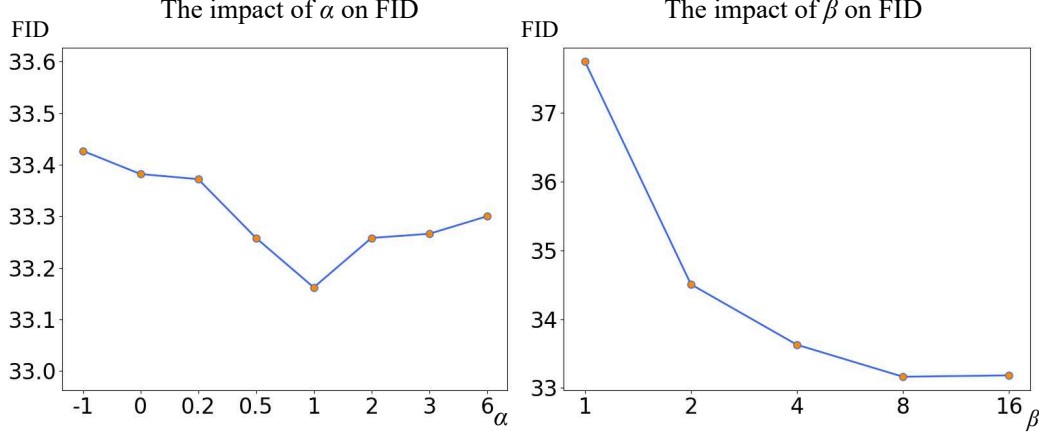

Figure A7: Performance plots (FID) of varying values of the hyper-parameters $\alpha$ (left) and $\beta$ (right) on the HE2IHC [61] dataset. Note that the labels for the horizontal axes are indicative and do not strictly follow the actual intervals between values.

As shown in Figure A7 (left), the best performance occurs at $\alpha = 1$, while extreme values ($\alpha = 0$ or 6) degrade performance. Notably, $\alpha = 0$, which is effectively the random patch selection, yields the second worst performance. In addition, we experiment with sampling more patches in high-frequency

Table A8: Evaluation of various virtual staining methods on two external validation datasets (and virtual staining tasks). For all methods, the models trained for the corresponding tasks in the main text are directly used here for inference without any tuning.

| Method | DISTS↓ | FID↓ | KID↓ | PSNR↑ | SSIM↑ |
|---|---|---|---|---|---|
| *Ext-HE2IHC* [61] | | | | | |
| Pix2pix [37] | 0.2923 | 147.02 | 0.1065 | 16.632±3.442 | 0.3250±0.126 |
| Pix2pixHD [83] | 0.2792 | 104.96 | 0.0597 | 16.361±3.390 | 0.3419±0.128 |
| CycleGAN [100] | 0.2823 | 149.61 | 0.1168 | **16.918±3.484** | 0.3410±0.117 |
| [61] | 0.2657 | 104.79 | 0.0796 | 15.708±2.234 | 0.3451±0.130 |
| SynDiff [56] | 0.3584 | 251.49 | 0.2500 | 16.700±3.356 | 0.3552±0.099 |
| StainFuser [39] | 0.2634 | 156.55 | 0.0932 | 16.131±3.336 | 0.3436±0.143 |
| D-VST | **0.2628** | **88.287** | **0.0396** | 16.713±3.598 | **0.3601±0.140** |
| *Ext-FFPE2HE* [34] | | | | | |
| Pix2pix [37] | 0.2066 | 44.919 | 0.0271 | 15.780±1.853 | 0.4580±0.076 |
| Pix2pixHD [83] | 0.1708 | 28.332 | 0.0139 | 17.726±1.457 | 0.4789±0.084 |
| CycleGAN [100] | 0.2913 | 99.343 | 0.0780 | 11.778±2.055 | 0.2774±0.125 |
| [34] | 0.1908 | 51.966 | 0.0342 | **18.975±1.404** | **0.5932±0.073** |
| SynDiff [56] | 0.3420 | 154.37 | 0.1344 | 6.8810±2.005 | 0.2219±0.114 |
| StainFuser [39] | 0.2153 | 71.887 | 0.0408 | 12.538±1.051 | 0.2326±0.093 |
| D-VST | **0.1521** | **27.488** | **0.0108** | 17.179±1.036 | 0.4591±0.075 |

regions by setting $\alpha = -1$. The performance is worse than that of random sampling ($\alpha = 0$) and our low-frequency-preferred sampling ($\alpha = 1$), validating that mosaic artifacts impact low-frequency regions more. We use $\alpha = 1$ for experiment comparison with other methods in the main text.

As for $\beta$, we study its impact with $\alpha$ fixed to 1. Figure A7 (right) shows that increasing the number of sampled patches rapidly improves the performance, which is reasonable, until the saturation at $\beta = 16$—with a similar performance to $\beta = 8$. Considering that $\beta = 16$ doubles the amount of computation of $\beta = 8$, we set $\beta = 8$ for performance comparison in the main text.

**Validate sampling probability design.** To validate our design of sampling probabilities in Eqn. (2), we use the medium $l$ of images as the threshold for low- and high-frequency patches. It turns out that our method samples 69% of patches in low-frequency squares versus 31% in high-frequency ones. These numbers indicate that the sampling probabilities work as designed.

**External validation.** To further evaluate the generalizability of our method, we conduct external validation on two datasets: (1) the external test set from [61], comprising 285 HE-IHC image pairs stained with SP3 [14] or CB11 [62] antibodies (Ext-HE2IHC); and (2) the external test set from [34], containing 1,398 FFPE-HE image pairs (Ext-FFPE2HE). It is worth noting that for the external validation, we directly use the models trained for the corresponding tasks in the main text without further tuning. As shown in Table A8, D-VST again achieves the best performance for the three perception-oriented metrics (DISTS, FID, and KID) on both external test datasets, and competitive performance for PSNR and SSIM (ranking top one to top three among all compared methods). These results are consistent with those presented in the main text, demonstrating D-VST's strong generalizability in cross-dye virtual staining of histopathology images.

**Additional comparison with existing methods.** In this section, we compare our D-VST with two additional state-of-the-art approaches to virtual staining, PSRVS [98] and DeepLIIF [23], on the HE2IHC [61] dataset. The former belongs to diffusion-based models, and the latter to GAN-based. The results are shown in Table A9. With the due caution that these two methods may not be fully optimized for this task, we can see that D-VST substantially outperforms PSRVS and DeepLIIF in DISTS, FID, and KID, and is comparable in PSNR and SSIM.

**Additional qualitative results.** We present additional qualitative tone control results on HE2IHC and FFPE2HE in Figure A9, under settings consistent with Table 1 and Figure 5 of the main text. We also show qualitative visual comparisons with other methods in Figure A10 and Figure A11 under the same settings. Figure A12 displays more qualitative comparisons between different sampling strategies

Table A9: Additional performance comparison with PSRVS [98] and DeepLIIF [23] on the HE2IHC dataset.

| Method | DISTS↓ | FID↓ | KID↓ | PSNR↑ | SSIM↑ |
|---|---|---|---|---|---|
| D-VST (ours) | **0.154** | **33.16** | **0.0055** | **18.11±3.874** | 0.407±0.136 |
| PSRVS | 0.422 | 230.3 | 0.2283 | 17.94±4.193 | 0.396±0.128 |
| DeepLiiF | 0.305 | 140.3 | 0.1379 | 17.96±3.713 | **0.418±0.129** |

for zero-shot virtual staining of large histopathology images at the resolution of 2048×2048 pixels. These results qualitatively demonstrate that D-VST can (1) precisely control the tones of the cross-dye virtually stained histopathology images without pathology leakage, (2) generate high-quality large histopathology images in an efficient manner, and (3) support virtual staining of ultra-high-resolution WSIs, validating the versatile generative capacity and generalization ability of D-VST.

**WSI validation.** As far as we know, no suitable WSI dataset is currently available for reliable quantitative evaluation at scale. Therefore, we mainly show the qualitative results in Figure A13, Figure A14, and Figure A15 for an observational study. Figure A13 shows ultra-high-resolution (16,384×16,384 pixels) HE2IHC virtual staining examples. Lastly, Figure A14 and Figure A15 showcase virtual staining results of WSIs (16,000×15,096 and 40,000×32,496 pixels) from Ext-FFPE2HE, each conditioned with two different target tones. We have asked two board-certified pathologists to blindly rank the WSIs virtually stained by our D-VST, a representative GAN-based method (pix2pixHD [83]), and two diffusion-based methods (SynDiff [56] and StainFuser [39]), considering image quality and pathology correctness. The evaluated WSIs include two HE2IHC and two FFPE2HE images, shown in in Figure A13, Figure A14, and Figure A15, respectively. The mean ranking is: D-VST (1.0), pix2pixHD (2.5), StainFuser(3.0), and SynDiff (3.5), demonstrating D-VST's superior virtual staining quality.

As for timing, we record the generation times for the WSI in Figure 1 of the main text (16,384×16,384 pixels; HE to IHC) using a representative GAN-based method (pix2pixHD [83]), two diffusion-based methods (StainFuser [39] and SynDiff [56]), and our D-VST. To avoid the unwanted mosaic artifacts, we implement the MultiDiffusion [3] sampling strategy for the compared methods, whereas D-VST uses its adaptive sampling strategy. As shown in Table A10, D-VST is orders of magnitude faster than the other diffusion-based methods and comparable to the GAN-based approach, highlighting its efficiency advantage for large WSI virtual staining over existing diffusion-based methods.

Table A10: Runtime comparison of Pix2pixHD [83], StainFuser [39], SynDiff [56], and D-VST for generating a WSI of 16,384×16,384 pixels.

| Method | Pix2pixHD | StainFuser | SynDiff | D-VST (ours) |
|---|---|---|---|---|
| Time (second) | 7,127 | 840,499 | 172,032 | 8,862 |

**Influence of fine-tuning variational autoencoder (VAE).** D-VST inherits the Stable Diffusion (SD) VAE [67] from PixArt-$\alpha$ [9], as it adopts the Diffusion Transformer (DiT) from PixArt-$\alpha$ as the denoising network. Our empirical evidence, both quantitative and qualitative, shows that D-VST with the SD VAE outperforms various GAN- and diffusion-based approaches on both virtual staining and downstream classification tasks. Thus, we argue that the SD VAE suffices for encoding and decoding of pathology images in D-VST, despite being pretrained on natural images and probably not being the optimal choice. We attribute this to SD VAE's strong encoding/decoding ability from large-scale, diverse training, and DiT's high compatibility with it. Notably, other works (e.g., HistDiST [25], StainFuser [39]) also successfully used VAEs pretrained on natural images for histopathology virtual staining.

We also experimented with fine-tuning the SD VAE on the HE2IHC dataset, but observed mixed results: similar PSNR/SSIM but worse DISTS, FID, and KID. We visualize the reconstruction error distributions in Figure A8. The reconstructed images show blurred cell membranes in lesion regions. As shown in the MSE maps, these lesion regions exhibit the highest intensity, indicating that the reconstructed images deviate most from the ground truth in these areas. We speculate that this

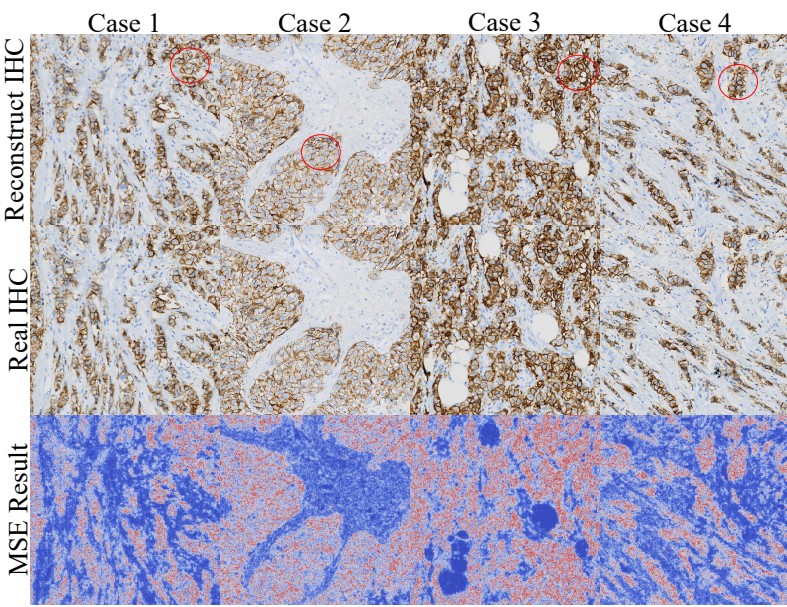

Figure A8: *Reconstruct IHC* denotes the generated images produced with the fine-tuned VAE, whereas *Real IHC* represents the corresponding ground-truth images. *MSE Result* refers to the pixel-wise mean squared error (MSE) maps between the reconstructed and real images, where regions with higher temperature indicate larger discrepancies. The red ellipses outline some areas where the reconstructed cell membranes in lesion regions appear noticeably blurred.

degradation is due to the limited training data, which prevented the VAE from learning pathological structures effectively; however, further validation with more data is needed.

**Impact of tone control image selection on the downstream task.** To evaluate the robustness of D-VST to tone control images, we conduct an additional set of experiments in which only a single IHC conditioning patch is used. Specifically, we randomly select one IHC image from the sub-test set as the sole tone conditioning patch and repeat the experiment three times with different selections, reporting the averaged results in Table A11. Otherwise, these experiments follow the same setting as Table 3 in the main text. As we can see, the performance with a single IHC conditioning patch is comparable to that obtained using multiple patches (originally reported in Table 3). This demonstrates that D-VST's downstream task performance is robust to the number of tone conditioning images, which we attribute to its effective disentanglement of tone and pathological conditions.

Table A11: Impact of tone control image selection on the downstream task.

| No. tone patches | ACC | F1 | Precision | Recall |
|---|---|---|---|---|
| Multiple (original) | 0.9417 | 0.9430 | 0.9470 | 0.9388 |
| Single (avg. over 3 runs) | $0.9415 \pm 0.0068$ | $0.9427 \pm 0.0064$ | $0.9483 \pm 0.0050$ | $0.9416 \pm 0.0068$ |

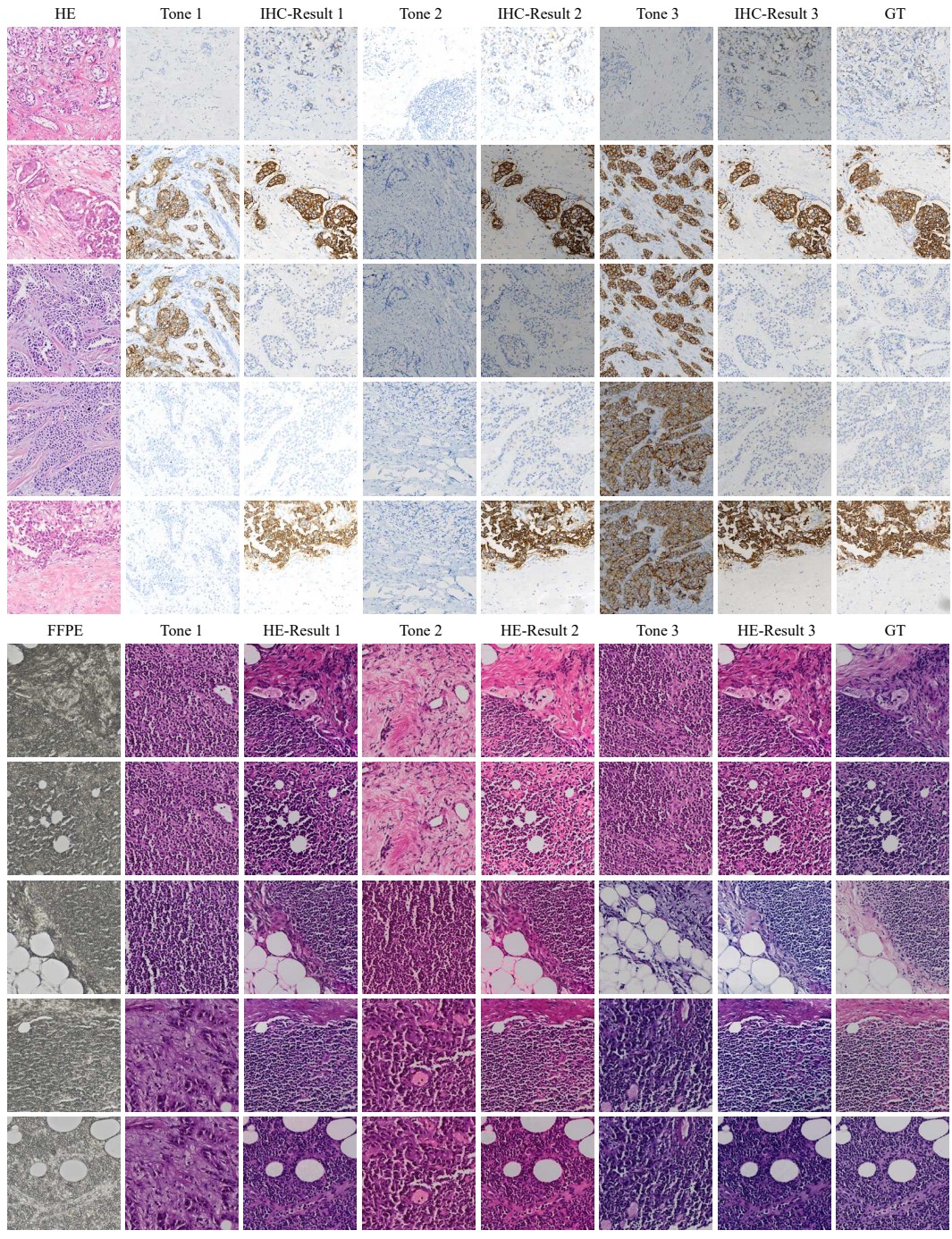

Figure A9: Additional qualitative results for tone control. Top: HE to IHC; and bottom: FFPE to HE. The images are virtually stained at the resolution of 1024×1024 pixels. GT: ground truth.

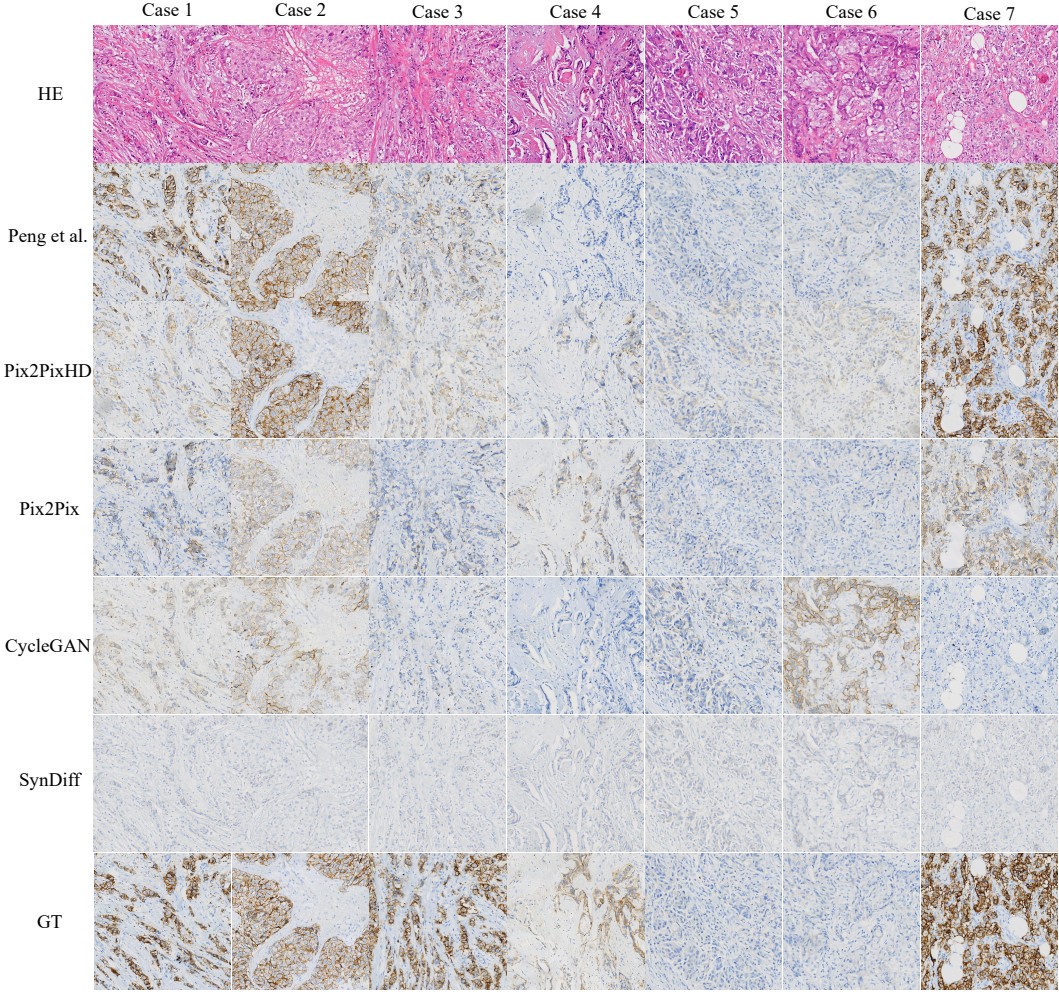

Figure A10: Qualitative comparison of HE to IHC virtual staining results by various methods that cannot control the tone of the virtually stained images, including Peng et al. [61], Pix2PixHD [83], Pix2Pix [37], CycleGAN [100], and SynDiff [56]. Note that the corresponding results by a few methods that control the tone of the re-stained images through image-based conditioning (including ours) are presented in Figure A11 for comparison. The images are virtually stained at the resolution of 1024×1024 pixels. GT: ground truth. By comparing the virtually stained images with the GT, we can observe clear Type I (false positive, meaning hallucinated cancerous status) and Type II (false negative, meaning hallucinated cancer-free status) errors for these methods.

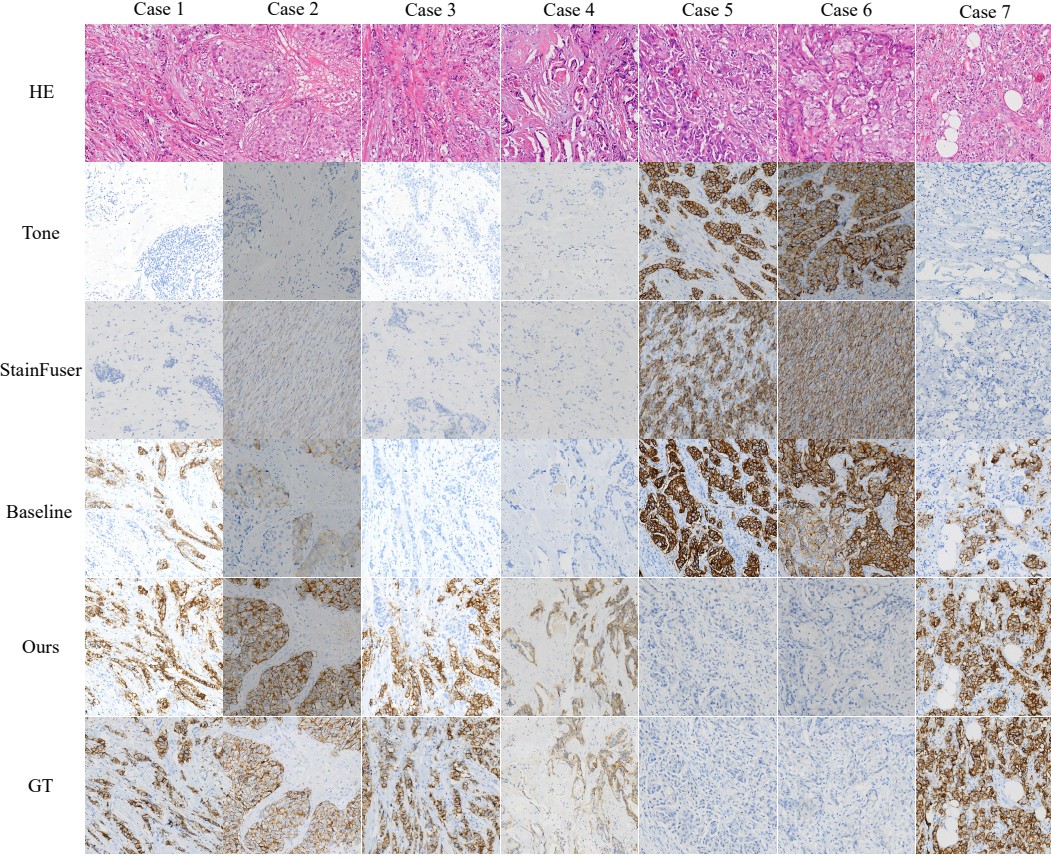

Figure A11: Qualitative comparison with other methods for HE to IHC virtual staining with tone control, including StainFuser [39], baseline (training the same network in Figure 3 (left) as our D-VST but without the proposed two-stage condition-decoupling curriculum learning). The "Tone" row displays the tone-conditioning images. Note that the corresponding results by methods that cannot control the tone of the re-stained images are shown in Figure A10 for comparison. The images are virtually stained at the resolution of $1024 \times 1024$ pixels. GT: ground truth. By comparing the virtually stained images with the GT, we can observe clear pathology leakages of Type I (false positive, meaning hallucinated cancerous status) and Type II (false negative, meaning hallucinated cancer-free status) errors for the compared methods. In contrast, the results of our D-VST align closely with the pathological statuses and distributions of the GT, while accurately reflecting the tones of the conditioning images.

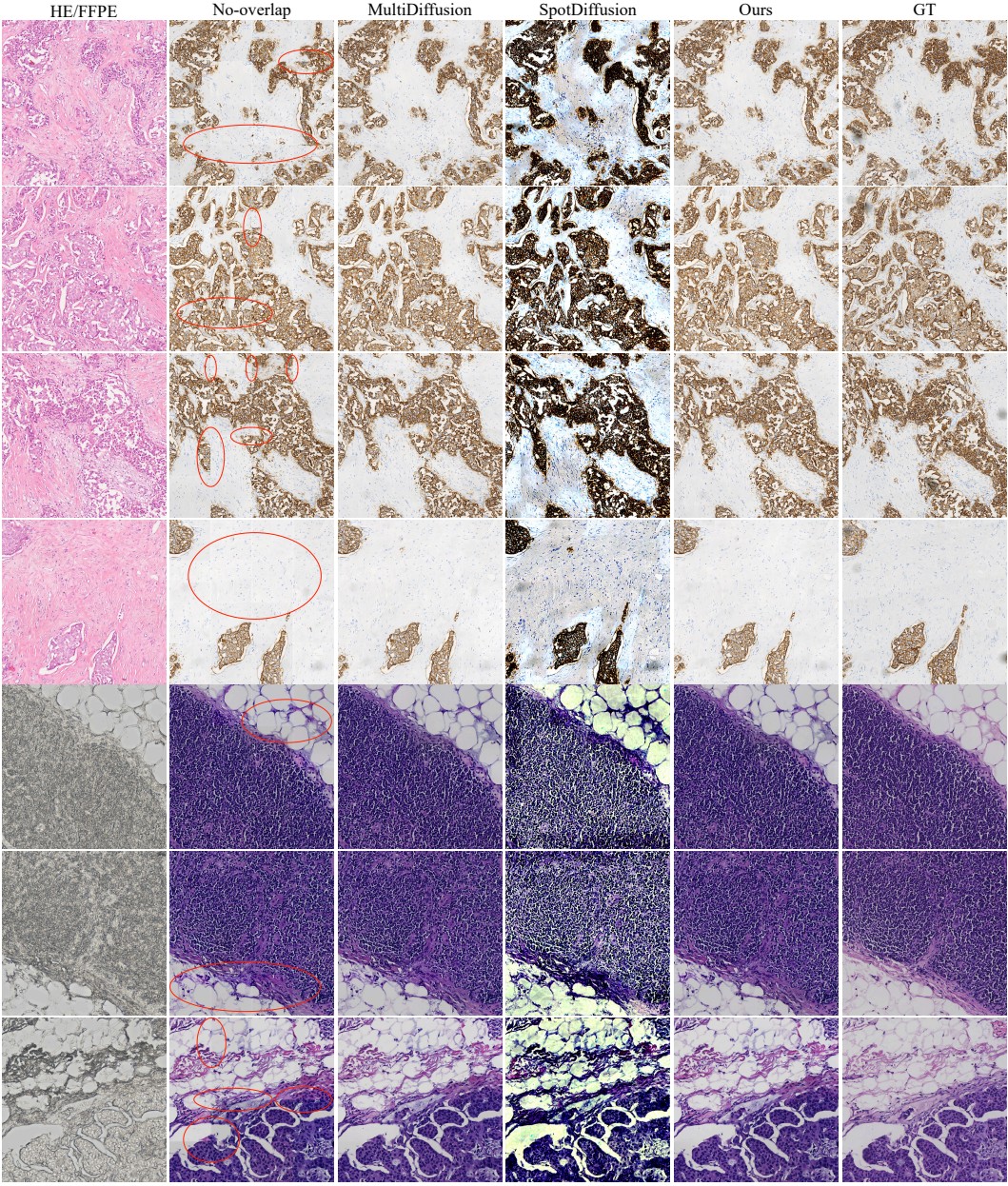

Figure A12: Comparison of sampling strategies for zero-shot virtual staining of large histopathology images (2048×2048 pixels). The red ellipses outline regions where the mosaic artifacts are prominent (best viewed zoomed in). GT: Ground truth.

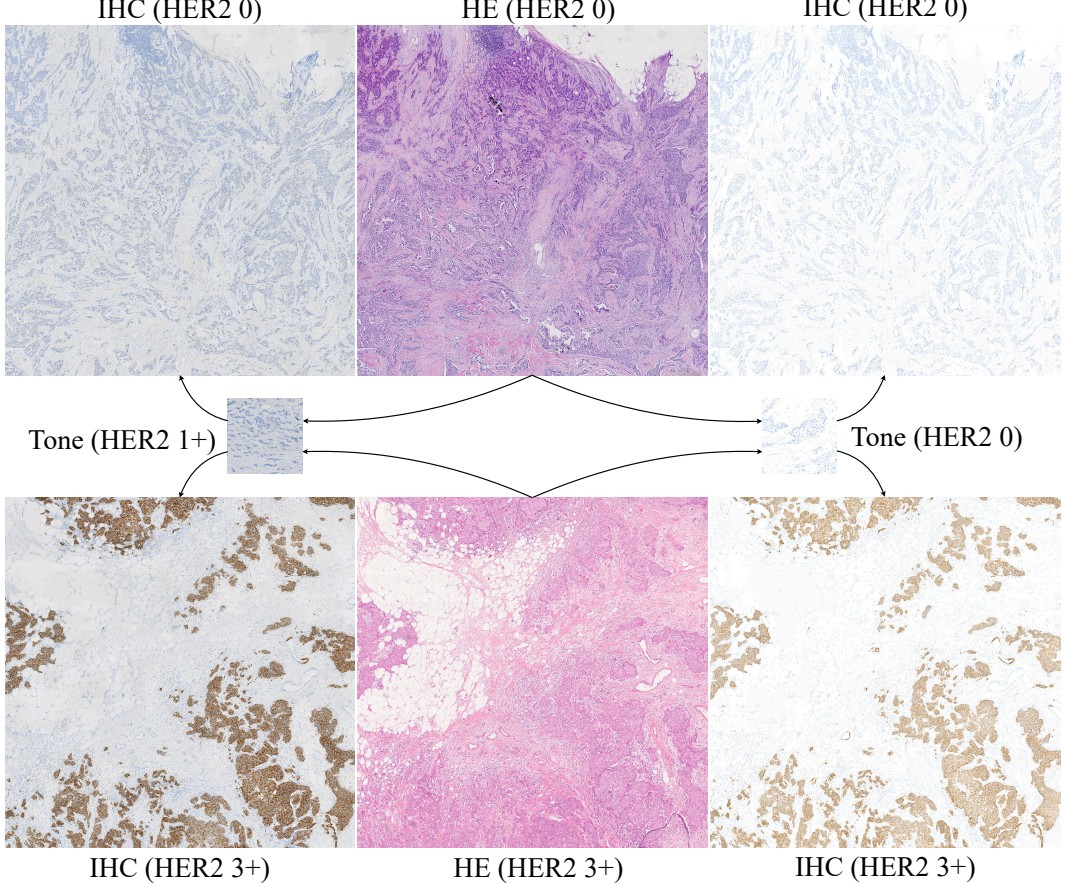

Figure A13: Examples of ultra-high-resolution (16,384×16,384 pixels) HE2IHC virtual staining. The central column shows the source-domain HE images, whereas the left and right columns show the virtually stained IHC images conditioned on two different tone-control images from the target domain. The top and bottom rows show two examples. HER2 scores: 0: no cancerous lesion, and 1+, 2+, and 3+: increasing severity of cancerous lesions.

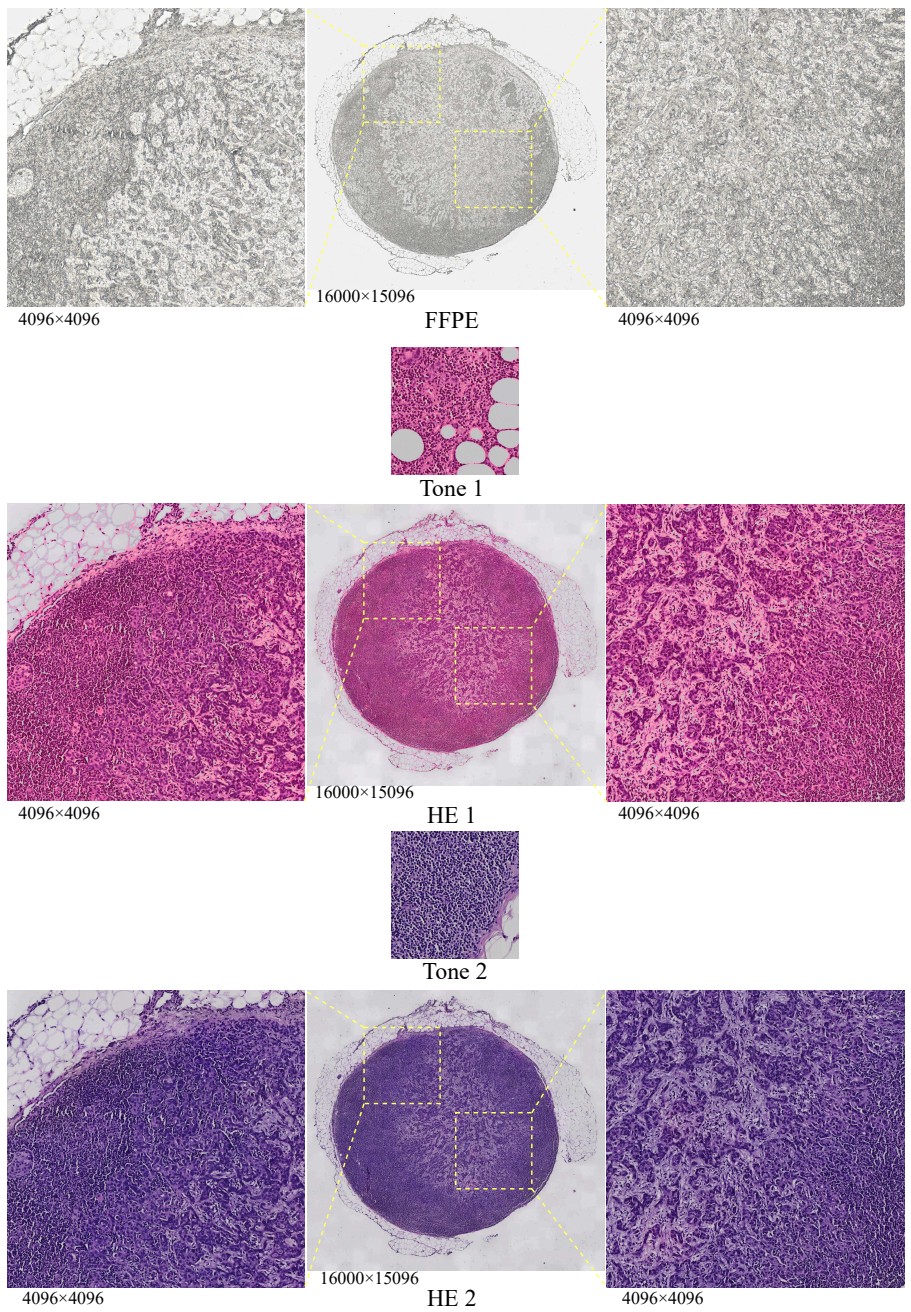

Figure A14: Virtual staining of an FFPE WSI in Ext-FFPE2HE, illustrated with two tone-conditioning images (Tone 1 and Tone 2) from the target domain. The virtual staining results corresponding to Tone 1 and Tone 2 are HE 1 and HE 2, respectively.

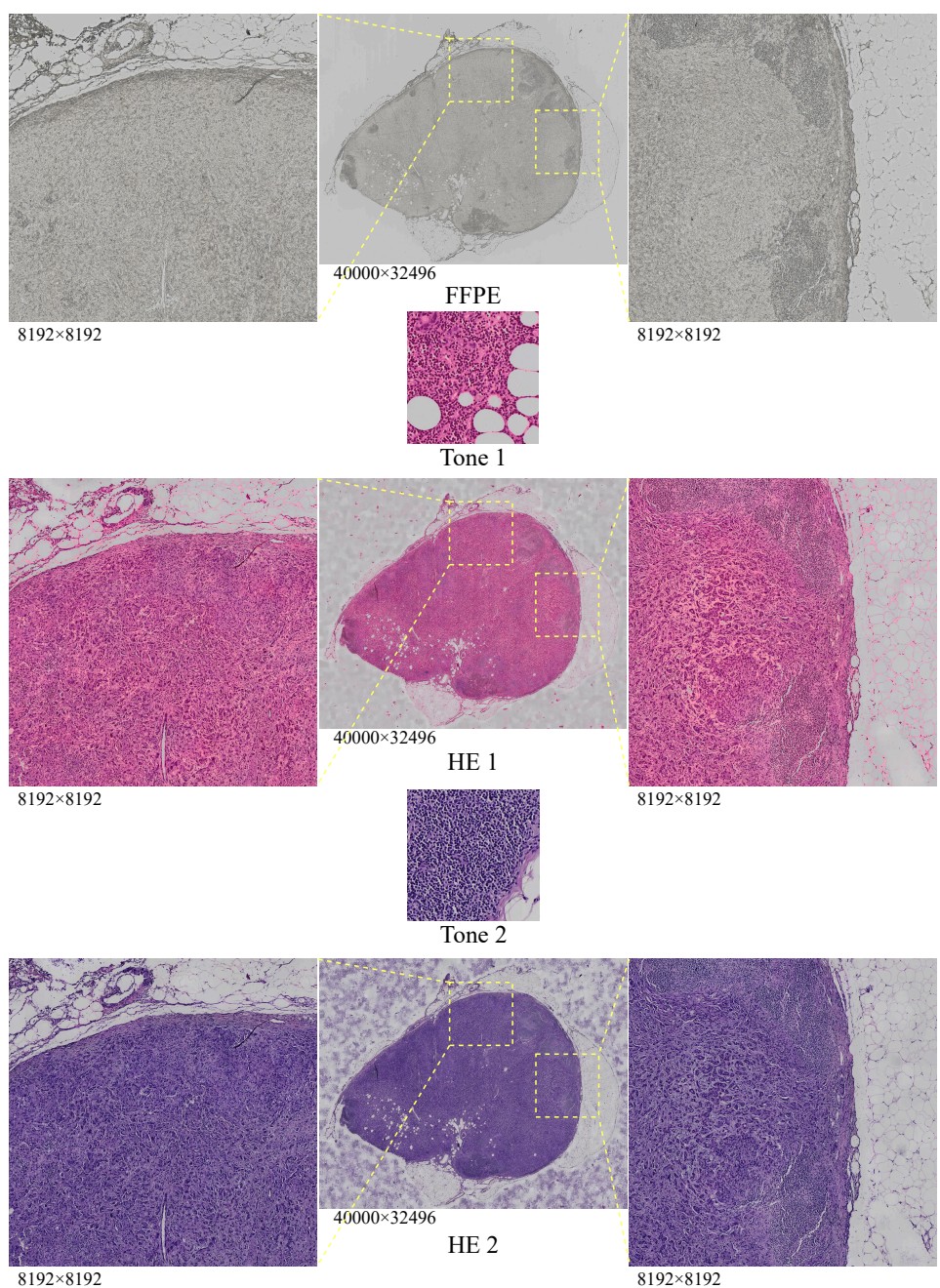

Figure A15: Virtual staining of an FFPE WSI in Ext-FFPE2HE, illustrated with two tone-conditioning images (Tone 1 and Tone 2) from the target domain. The virtual staining results corresponding to Tone 1 and Tone 2 are HE 1 and HE 2, respectively.

