# OpenReview forum: "D-VST: Diffusion Transformer for Pathology-Correct Tone-Controllable Cross-Dye Virtual Staining of Whole Slide Images"
_NeurIPS.cc/2025/Conference — NeurIPS 2025 poster_

### Official Review · Reviewer_XhuF · 2025-06-13

**Clarity:** 2
**Significance:** 3
**Originality:** 3
**Rating:** 5
**Confidence:** 5

**Summary:**

This paper presents D-VST (Diffusion Virtual Staining Transformer), a new diffusion-based method for virtual staining of histopathology images, with a focus on accurate pathology preservation and flexible tone control. Traditional methods often suffer from pathology leakage, where incorrect staining occurs due to entanglement of pathology and tone information. To address this, the authors propose:

1. A dual-encoder architecture (pathology and tone encoders) that separates content and style in the staining process.
2. A two-stage curriculum learning strategy to effectively train the model on these disentangled conditions.
3. A frequency-aware adaptive patch sampling technique for efficient and scalable inference on ultra-high-resolution whole slide images, enabling zero-shot generalization.

The method demonstrates strong performance across three stain translation tasks, outperforming existing GAN-based and diffusion-based approaches in both visual quality and pathological fidelity.

**Questions:**

1. Clarification and Supporting Evidence for Figure 1 Claims

Figure 1 highlights several core advantages of D-VST, such as being “Fast,” “Pathology-Correct,” and free of “Mosaic Artifacts.” However, these claims are not directly supported within the figure.

Suggestion: Please include visual examples (e.g., zoom-in patches with and without mosaic artifacts) and/or quantitative metrics that demonstrate the absence of artifacts and the correctness of pathology preservation. While including some visual or metric-based evidence in Figure 1 would be ideal, alternatively, you could explicitly reference other figures, tables, or sections in the paper that substantiate these claims. For example, show where “no mosaic artifact” is evidenced, or where “fast” is quantified.

Evaluation impact: Providing this evidence would significantly improve the clarity and credibility of the figure, and would strengthen the justification for your claims.

2. Runtime and Efficiency Comparison Missing

The method is described as fast, but no runtime or efficiency comparisons are provided.

Suggestion: Please provide a runtime or inference time comparison with baseline models (e.g., GAN-based and other diffusion-based methods), ideally on WSIs and/or standard image sizes.

Evaluation impact: Demonstrating competitive or superior efficiency would support real-world deployment claims and positively influence the significance of the work.

**Ethical Concerns:**

["NO or VERY MINOR ethics concerns only"]

**Final Justification:**

Thanks to the authors for their comprehensive response, which resolved my concerns. I have accordingly increased my score.

**Limitations:**

No. The authors have addressed several important limitations. To further strengthen the discussion, it would be helpful to acknowledge the reliance on cross-dye paired training data, which may be challenging to obtain in practice. Highlighting this would enhance the clarity and practical relevance of the work.

**Paper Formatting Concerns:**

No major formatting issues noticed.

**Quality:**

3

**Strengths And Weaknesses:**

Strengths:

1. First Diffusion-Based Model with Tone Control for Cross-Dye Staining

The proposed Diffusion Virtual Staining Transformer (D-VST) is the first diffusion-based model to support versatile tone control in cross-dye virtual staining, marking a meaningful advance in the virtual staining literature.

2. Pathology and Tone Representations Decoupling via Curriculum Learning

D-VST introduces a two-stage curriculum learning strategy to disentangle pathology and tone representations, effectively addressing the critical issue of pathology leakage, where pathological and non-pathological regions are misrepresented during virtual staining.

3. Frequency-Aware Adaptive Sampling for High-Quality Zero-Shot Virtual Staining for WSIs

To mitigate mosaic artifacts and enhance zero-shot generation quality on whole slide images (WSIs), the authors introduce a frequency-aware adaptive patch sampling strategy. By prioritizing sampling in low-frequency regions, where artifacts are more likely, this method enables efficient and high-fidelity inference on billion-pixel WSIs.

4. Comprehensive Experimental Evaluation

The method is validated on three cross-dye staining tasks involving four stain types, with additional downstream classification tasks and ablation studies, demonstrating strong performance against recent approaches.

Weaknesses:

1. Insufficient Explanation of HER2 Labels in Figure 1

Figure 1 includes references to HER2 0, 1+, 2+, 3+, but lacks explanation of what these labels mean clinically. This omission reduces clarity for readers unfamiliar with HER2 scoring systems.

2. Evidence for Key Claims Should Be Included in Figure 1

The caption of Figure 1 claims the method is "Fast", "Pathology-Correct", and “No Mosaic Artifacts”, but no visual or quantitative evidence is provided to substantiate these assertions in this figure.

3. Missing Runtime Comparison

While the paper asserts that the method is computationally efficient, there is no runtime or inference speed comparison with other models.

4. Requirement for Cross-Stain Paired Training Data

D-VST relies on paired cross-stain training data, which can be challenging and costly to obtain in real-world medical imaging workflows. This reliance limits the method’s scalability and broader applicability, yet the paper does not discuss this practical constraint or propose solutions for unpaired or weakly supervised settings.

---

> ### Author Rebuttal · Authors · 2025-07-31
>
> > **C1 (W1)**: Insufficient Explanation of HER2 Labels in Figure 1: Figure 1 includes references to HER2 0, 1+, 2+, 3+, but lacks explanation of what these labels mean clinically. This omission reduces clarity for readers unfamiliar with HER2 scoring systems.
>
> **A1**: We appreciate the reviewer's valuable comment.
>
> To improve clarity for readers unfamiliar with HER2 scoring systems, we will **add an explanation of the HER2 scores in the caption of Figure 1**: 0 indicates no cancerous lesion, while 1+, 2+, and 3+ represent increasing severity of cancerous lesions.
>
> > **C2 (W2)**: Evidence for Key Claims Should Be Included in Figure 1: The caption of Figure 1 claims the method is "Fast", "Pathology-Correct", and “No Mosaic Artifacts”, but no visual or quantitative evidence is provided to substantiate these assertions in this figure.
> >
> > **(Q1)**: Clarification and Supporting Evidence for Figure 1 Claims: Figure 1 highlights several core advantages of D-VST, such as being “Fast,” “Pathology-Correct,” and free of “Mosaic Artifacts.” However, these claims are not directly supported within the figure.
> >
> > Suggestion: Please include visual examples (e.g., zoom-in patches with and without mosaic artifacts) and/or quantitative metrics that demonstrate the absence of artifacts and the correctness of pathology preservation. While including some visual or metric-based evidence in Figure 1 would be ideal, alternatively, you could explicitly reference other figures, tables, or sections in the paper that substantiate these claims. For example, show where “no mosaic artifact” is evidenced, or where “fast” is quantified.
> >
> > Evaluation impact: Providing this evidence would significantly improve the clarity and credibility of the figure, and would strengthen the justification for your claims.
>
> **A2**: We thank the reviewer for the detailed, helpful suggestions.
>
> To address these points, we will revise Figure 1 as follows:
>
> - **Fast**: We will reference the inference time comparison table (see our response to the following comment), which shows that our D-VST re-stains the large WSI image in Figure 1 orders of magnitude faster than representative diffusion-based approaches.
> - **No Mosaic Artifacts**: We will add zoomed-in patch views to demonstrate the absence of mosaic artifacts. Also, we will refer to Figure 6 and Figure A5 (Appendix), which contrast the mosaic-artifact-free results by D-VST with those with noticeable mosaic artifacts by the No-overlap method.
> - **Pathology-Correct**: The zoomed-in patch views will also illustrate pathology correctness. Additionally, we will reference 1) Figures A3 and A4 (Appendix), which qualitatively contrast our pathology-correct virtual staining results with results by other methods; and 2) Table 3, which quantitatively compares the pathology accuracy of virtually stained images by various approaches, with a downstream pathology image classification task. Both the qualitative and quantitative results show that our method outperforms the compared ones.
>
> We believe these changes will make Figure 1 clearer and better supported, strengthening the justification for our claims.
>
> > **C3 (W3)**: Missing Runtime Comparison: While the paper asserts that the method is computationally efficient, there is no runtime or inference speed comparison with other models.
>
> > **(Q2)**: Runtime and Efficiency Comparison Missing: The method is described as fast, but no runtime or efficiency comparisons are provided.
> >
> > Suggestion: Please provide a runtime or inference time comparison with baseline models (e.g., GAN-based and other diffusion-based methods), ideally on WSIs and/or standard image sizes.
> >
> > Evaluation impact: Demonstrating competitive or superior efficiency would support real-world deployment claims and positively influence the significance of the work.
>
> **A3**: Thanks for the constructive suggestion.
>
> Accordingly, we have recorded the generation times for the WSIs in Figure 1 (16,384×16,384 pixels; HE to IHC) using a representative GAN-based method (pix2pixHD), two diffusion-based methods (StainFuser and SynDiff), and our D-VST.
> To avoid the unwanted mosaic artifacts, we implement the MultiDiffusion [3] sampling strategy for the compared methods, while D-VST uses its adaptive sampling strategy.
> The results are:
> | Method | Pix2pixHD | StainFuser | SynDiff | D-VST |
> |-------|-------|-------|-------|-------|
> | Time (second) | 7,127 | 840,499 | 172,032 | 8,862 |
>
> As we can see, **our D-VST is orders of magnitude faster than the other diffusion-based methods and comparable to the GAN-based approach**, highlighting its efficiency advantage for large WSI virtual staining over existing diffusion-based methods.
>
> > **C4 (W4)**: Requirement for Cross-Stain Paired Training Data: D-VST relies on paired cross-stain training data, which can be challenging and costly to obtain in real-world medical imaging workflows. This reliance limits the method’s scalability and broader applicability, yet the paper does not discuss this practical constraint or propose solutions for unpaired or weakly supervised settings.
> >
> > **(Limitations)**: No. The authors have addressed several important limitations. To further strengthen the discussion, it would be helpful to acknowledge the reliance on cross-dye paired training data, which may be challenging to obtain in practice. Highlighting this would enhance the clarity and practical relevance of the work.
>
> **A4**: We thank the reviewer for acknowledging our efforts in addressing several important limitations, and more importantly, for pointing out another limitation that should be discussed.
>
> We acknowledge that obtaining paired cross-stain training data can be challenging and costly in real-world workflows, which may limit the scalability and applicability of D-VST.
> While such data can improve virtual staining performance by paired correspondence, unpaired data is substantially more scalable due to orders of magnitude larger amounts.
> In future work, we plan to **explore benefiting from both the scalability of unpaired data and the quality of paired data via a combination of D-VST and approaches like CycleGAN-Turbo [A], which enable diffusion models to learn from unpaired data.**
> We will add this discussion to the final manuscript.
>
> [A] Parmar, Gaurav, et al. "One-step image translation with text-to-image models." arXiv preprint arXiv:2403.12036 (2024).

---

> > ### Comment · Reviewer_XhuF · 2025-08-08
> >
> > Thank you for clarifying my questions. I have accordingly increased my score.

---

> > > ### Author Response · Authors · 2025-08-08
> > >
> > > ​We are delighted to know that our response has clarified your questions, and sincerely thank you for your thoughtful review, which helps strengthen our paper. We are truly grateful for your positive feedback.

---

### Official Review · Reviewer_yaZj · 2025-06-30

**Clarity:** 3
**Significance:** 3
**Originality:** 3
**Rating:** 5
**Confidence:** 2

**Summary:**

This work proposes a method for integrating tone control for virtual staining using diffusion transformer models. This integration is done to reduce false virtual staining of regions in tissue images (e.g., staining of non-cancer regions as cancer regions). The method is evaluated on several datasets and virtual staining tasks.

**Questions:**

Page 7 (lines 224-225): “use a random patch from the same WSI but not overlapping with the target image for tone conditioning”. Per this sentence, it sounds like a separate patch is selected for a source image. Is this realistic? In a real setting, there may not be more than one target image (e.g., an IHC image in case of HE-to-IHC image generation). Clarification is needed for how tone conditioning patches are selected during the training and test phases in the experiments.

Page 8 (lines 274-276): “Next, for each HE image in the sub-test set, we randomly select an IHC image in the sub-test set that is not paired with the specific HE image as the tone condition for virtual staining.” Selecting a different tone condition image for each HE image may affect the experimental results. In a real application setting, there may not be multiple IHC images. Please clarify if multiple IHC patches were selected for tone conditioning in the experiments. If so, a set of experiments should be executed in which only one IHC conditioning patch is selected.

**Ethical Concerns:**

["NO or VERY MINOR ethics concerns only"]

**Final Justification:**

The authors have provided clarifications to my questions.

**Limitations:**

.

**Quality:**

3

**Strengths And Weaknesses:**

The use of tone control to improve virtual staining performance is different and interesting. The proposed approach implements a two-stage training strategy to incorporate pathology and tone information. It also employs an adaptive patch-sampling technique to improve image generation quality in whole slide images.

Experimental evaluation using several datasets and virtual staining tasks (HE-to-IHC, FFPE-to-HE and HE-to-mIHC) shows performance improvements by the proposed method in several metrics (FID, KID, SSIM, etc) compared with other methods.

The experimental evaluation also includes a downstream image classification task. This is a useful experiment as it shows the performance improvements are not just in image generation metrics (KID, FID, etc), but also have discernable impact in pathology focused tasks.

---

> ### Author Rebuttal · Authors · 2025-07-31
>
> > **Q1**: Page 7 (lines 224-225): “use a random patch from the same WSI but not overlapping with the target image for tone conditioning”. Per this sentence, it sounds like a separate patch is selected for a source image. Is this realistic? In a real setting, there may not be more than one target image (e.g., an IHC image in case of HE-to-IHC image generation). Clarification is needed for how tone conditioning patches are selected during the training and test phases in the experiments.
>
> **A1**: We appreciate the reviewer's insightful comments.
>
> To clarify, a separate patch is indeed selected for each source image.
> **For our D-VST, the tone image can be any image from the target domain with the desired tone**.
> In our experimental scenarios, the images having tones closest to the target image happen to be the patches from the same WSI.
> Therefore, we selected a random patch that does not overlap with the target image (to prevent ground truth information leak) to provide the desired tone, but this is not a requirement for practical use.
> This setting is consistently used for model training and testing for results in Table 1, as described in Lines 175-178 and 224-226.
>
> Meanwhile, we acknowledge the reviewer's concern that there may not be more than one target-domain image in practice, forbidding the use of a separate tone conditioning image for each source image.
> In such cases, **D-VST can use the single available target-domain image for tone control, resulting in virtually stained images with tones similar to that image and correct pathology**.
> We have conducted additional experiments using one tone-conditioning image for all source images (see response to Q2).
> **The results show that D-VST remains effective with only one tone conditioning image and is not sensitive to the number of tone conditioning images.**
>
> > **Q2**: Page 8 (lines 274-276): “Next, for each HE image in the sub-test set, we randomly select an IHC image in the sub-test set that is not paired with the specific HE image as the tone condition for virtual staining.” Selecting a different tone condition image for each HE image may affect the experimental results. In a real application setting, there may not be multiple IHC images. Please clarify if multiple IHC patches were selected for tone conditioning in the experiments. If so, a set of experiments should be executed in which only one IHC conditioning patch is selected.
>
> **A2**: Thank you for the valuable suggestion.
>
> Above all, we clarify that multiple IHC patches were used for tone conditioning in these experiments, as our primary target was to demonstrate that our D-VST can achieve pathology-leakage-free virtual staining with precise tone control.
> Therefore, selecting an IHC image that is different from the real one for each HE image already suffices for the experiments, without the need for constraining the total number of selected ones.
>
> However, we acknowledge the reviewer's concern that in a real application setting, there may not be multiple IHC images for tone control.
> To address this concern, we have **conducted a set of additional experiments in which only one IHC conditioning patch is selected**.
> Specifically, we randomly select an IHC image from the sub-test set as the only tone conditioning patch and repeat the experiment three times with different images selected, reporting the average results below:
> | No. tone patches | ACC | F1 | Precision | Recall |
> | ------ | ------ | ------ | ------ | ------ |
> | Multiple (original) | 0.9417 | 0.9430 | 0.9470 | 0.9388 |
> | Single (avg. over 3 runs) | 0.9415±0.0068 | 0.9427±0.0064 | 0.9483±0.0050 | 0.9416±0.0068 |
>
> As shown, **the performance with a single IHC conditioning patch is similar to that with multiple patches** (originally reported in our paper).
> This demonstrates that D-VST's performance on the downstream task is robust to the number of tone conditioning images, which we attribute to its effective decoupling of tone and pathology conditions.

---

> > ### Comment · Reviewer_yaZj · 2025-08-03
> > **Authors' responses**
> >
> > Thank you for clarifying the questions.

---

> > > ### Author Response · Authors · 2025-08-05
> > >
> > > Thank you very much for your thoughtful review and for appreciating our work. We truly appreciate your positive feedback.

---

### Official Review · Reviewer_843m · 2025-07-01

**Clarity:** 4
**Significance:** 4
**Originality:** 3
**Rating:** 5
**Confidence:** 4

**Summary:**

The paper presents a diffusion transformer for virtual staining, e.g IHC staining from HE images, or HE staining from FFPE (unstained tissue). It has two main contributions: 1) it uses a two-stage curriculum learning scheme to separate tone and pathology features, solving the pathology leakage problem, and ii) it uses a frequency-aware adaptive patch sampling strategy to mitigate mosaic artefacts during high-resolution inference. They evaluate their model on three datasets and show improved or competitive performance compared to state-of-the-art virtual staining methods.

**Questions:**

1. According to Session: Datasets and Evalaution Metrics (line 218-265), authors trained three models, one for each dataset and evaluated the model on the testing split of the corresponding dataset. This already raised questions on the model generalisation. As for clinical usage, a model need to be evaluated on an independent patient cohort.
2. Another question is for tone image selection. It is written that tone image is “ a random patch from the same WSI but not overlapping  with the target image for tone conditioning.” For model inference, do users need to select one tone image for the same WSI of a different stain? That leads to a practical problem, if users have the stained the same image with a different stain, why do they need the generated one? Is it possible to select the tone image from the training set.
3. There is rapid development for pathology foundation model recently,  why not use pathology foundation models trained on multi stains as conditioning like UNI, https://huggingface.co/MahmoodLab/UNI2-h
4. Ablation study on the autoencoder size would also be interesting, since authors only use a small and lightweight one.
5. Ref [49] is a very relevant work as it does something similar but on TMAs and using GNN. Could authors discuss their methodological differences?

**Ethical Concerns:**

["NO or VERY MINOR ethics concerns only"]

**Final Justification:**

Authors have addressed my remaining concerns and I will keep my original accept decision.

**Limitations:**

Maybe indeed authors can add a few sentence about the status of their model and potential limitations of using their model in clinical settings.

**Paper Formatting Concerns:**

did not find any

**Quality:**

4

**Strengths And Weaknesses:**

Strengths:

1: The paper is well written with clearly explained motivation, methodology and experimental design.

2. The methodology is logical and well-designed. Although the essential concept like disentanglement of pathology features and tones, is not necessarily new (e.g. used in histology stain normalisation/augmentation), but the design of the presented model architecture is novel, as far as I can see.

3. The experimental evaluation is also comprehensive, covering three different datasets. Particularly, the evaluation of downstream HER classification and scoring provides quantitative evidence of the proposed method over other methods in pathology leakage prevention (Table 3 back up qualitative results in Fig. 2) .

4. Ablation studies show that two-stage curriculum learning is indeed the key of pathology structure preservation in terms of prediction accuracy of HER classification (from around 0.77 to 0.94).

Weakness:
1. The theoretical contribution of the paper is limited as it is pretty much a very applied paper (not necessarily a weakness of the paper itself, but theoretical contribution is something I like for NeurIPS conference)
2. Sampling strategy seems only marginal better (according to Table 1). Is it worth the computational cost?
3. Not entirely clear about model generalisation (see Questions for details)

---

> ### Author Rebuttal · Authors · 2025-07-31
>
> > **C1 (W1)**: The theoretical contribution of the paper is limited as it is pretty much a very applied paper (not necessarily a weakness of the paper itself, but theoretical contribution is something I like for NeurIPS conference)
>
> **A1**: We thank the reviewer for **recognizing our significant methodological contributions**, including well-designed methodology, novel model architecture, and comprehensive experimental evaluation.
> Thus, we respectfully but firmly argue that our focus on applied aspects does not diminish the impact of our work, especially considering **its high significance in the pathology AI sub-area of NeurIPS**.
> Meanwhile, we provided **theoretical motivations and principles for our methodology design, supported by empirical evidence**.
>
> Most notably, the **two-stage curriculum learning** is the key to pathology structure preservation, as the reviewer noted.
> Its theoretical motivation is detailed in Lines 147-152: "... in Figure 2, **training with mingled pathology and tone controls often causes pathological status to leak through tone conditioning**, ... because **the model confuses the purposes of the conditioning images and mistakes the tone-conditioning image for the source of the pathology status**. Thus, **the key to effectively preventing pathology leakage is to decouple pathology and tone conditions by making the model learn both control signals precisely. For this purpose, we design a novel two-stage curriculum learning [4] scheme.**"
> In the first stage, the model learns to effectively extract and utilize the pathology information in the pathology-conditioning image.
> Then, in the second stage, the tone condition is introduced for more complex, joint learning of pathology and tone conditions.
> **This process enables effective decoupling of pathology and tone conditions**.
> Our ablation study in Table 4 shows that **removing the two-stage learning (w/o Curriculum) leads to the largest performance drops** in 3 of the 4 classification metrics, confirming our motivation and design principle.
>
> > **C2 (W2)**: Sampling strategy seems only marginal better (according to Table 1). Is it worth the computational cost?
>
> **A2**: Thank you for the insightful comment.
>
> We believe the reviewer referred to Table 2, which compares sampling strategies.
>
> Compared with no-overlap sliding windows, our strategy increases computational cost with similar PSNR, SSIM, and DISTS.
> However, it **substantially improves the perceptual metrics (FID and KID)**, e.g., FID on HE2IHC improves from 37.748 to 33.162.
> In addition, **no-overlap exhibits notable mosaic artifacts (Figure 6 and Appendix Figure A5), which cannot be effectively captured by the standard metrics but are visually significant**.
> In contrast, **our strategy eliminates these artifacts visually**.
> Hence, the quantitative and qualitative improvements justify the computational cost.
>
> Compared to MultiDiffusion [3], our strategy **achieves comparable (if not better) performance with only 1/8 computational cost, offering a worthy balance between quality and computational efficiency**.
>
> > **C3 (W3)**: Not entirely clear about model generalisation (see Questions for details)
>
> > **Q1**: According to Session: Datasets and Evalaution Metrics (line 218-265), authors trained three models, one for each dataset and evaluated the model on the testing split of the corresponding dataset. This already raised questions on the model generalisation. As for clinical usage, a model need to be evaluated on an independent patient cohort.
>
> **A3**: We appreciate the reviewer's insightful comments.
>
> The three datasets involve distinct source (HE/FFPE) and target domains (HE/IHC/mIHC), causing significant challenges to training a unified model for all three tasks.
> Thus, we followed most existing literature in training a separate model for each task, for agile method development and effective proof of concept, but plan to explore unified models in future work.
>
> Meanwhile, we agree that for clinical usage, a model needs to be evaluated on independent patient cohorts.
> We additionally validated our D-VST on two external validation datasets of HE2IHC and FFPE2HE, i.e., **the same tasks but different patient cohorts** (Appendix Lines 59-68).
> Notably, we directly evaluated the models on the external datasets without further tuning, to assess generalization.
> Appendix Table A3 shows that **D-VST again achieves the best performance for the perception-oriented metrics (DISTS, FID, and KID) on both external datasets**, and competitive performance for PSNR and SSIM.
> These results are consistent with those in the main text, demonstrating D-VST's **strong generalizability across independent patient cohorts**.
>
> > **C4 (Q2)**: Another question is for tone image selection. It is written that tone image is “a random patch from the same WSI but not overlapping with the target image for tone conditioning.” For model inference, do users need to select one tone image for the same WSI of a different stain? That leads to a practical problem, if users have the stained the same image with a different stain, why do they need the generated one? Is it possible to select the tone image from the training set.
>
> **A4**: Thank  you for the insightful questions.
>
> We would like to clarify that, for model inference, **any image from the target domain with the desired tone can be used**.
> In our experimental scenarios, the images having tones closest to the target image happen to be the patches from the same WSI.
> Therefore, we selected a random patch that does not overlap with the target image (to prevent ground truth information leak) to provide the desired tone, but this is not a requirement for practical use.
> Also, it is **feasible to select the tone image from the training set**.
> Intuitively, selecting images with tones closer to the target images would lead to better evaluation metrics.
>
> > **C5 (Q3)**: There is rapid development for pathology foundation model recently, why not use pathology foundation models trained on multi stains as conditioning like UNI
>
> **A5**: We appreciate the reviewer's constructive comment.
>
> Accordingly, we have experimented with replacing OpenAI-CLIP [52] with UNI in our D-VST.
> | Model | DISTS↓ | FID↓ | KID↓ | PSNR↑ | SSIM↑ |
> |-------|-------|-------|-------|-------|-------|
> | OpenAI-CLIP | 0.154 | 33.16 | 0.0055 | 18.11±3.874 | 0.407±0.136 |
> | UNI | 0.157 | 33.13 | 0.0048 | 18.03±3.842 | 0.404±0.130 |
>
> The above results on the HE2IHC dataset [51] show that **the performance is comparable between OpenAI-CLIP and UNI**.
> We believe that while UNI offers strong pathological feature extraction, OpenAI-CLIP suffices for capturing tone information in D-VST.
> Hence, **pathology foundation models trained on multi-stains like UNI are also excellent choices** for the proposed D-VST framework.
>
> > **C6 (Q4)**: Ablation study on the autoencoder size would also be interesting, since authors only use a small and lightweight one.
>
> **A6**: We thank the reviewer for the insightful comment.
>
> D-VST inherits the Stable Diffusion (SD) variational autoencoder (VAE) [56] from PixArt-α [8], as it adopts the Diffusion Transformer (DiT) from PixArt-α as the denoising network.
> **Our empirical evidence, both quantitative and qualitative**, shows that D-VST with the SD VAE outperforms various GAN- and diffusion-based approaches on both virtual staining and downstream classification tasks.
> This **suggests that the SD VAE suffices for encoding and decoding of pathology images *in D-VST*, despite being lightweight**.
> We attribute this to SD VAE’s strong encoding/decoding ability from large-scale, diverse training, and DiT's high compatibility with it.
>
> However, we agree with the reviewer that it is interesting to explore if enhancing the VAE, e.g., increasing its size, would bring further performance improvement.
> In a related attempt (Appendix Lines 91-95), we fine-tuned the SD VAE on our experimental data but observed mixed results (similar PSNR and SSIM with poorer DISTS, FID, and KID).
> We conjecture this might be due to the limited training data that prevented the VAE from learning pathological structures well, but further validation with more data is needed.
> In future work, **we plan to explore larger VAEs following the reviewer's suggestion**.
>
> > **C7 (Q5)**: Ref [49] is a very relevant work as it does something similar but on TMAs and using GNN. Could authors discuss their methodological differences?
>
> **A7**: We appreciate the reviewer's excellent suggestion.
>
> Key methodological differences include:
>
> - [49] uses GANs, while D-VST uses diffusion models;
> - [49] does not require paired data for training; D-VST does (further discussed in our response to the following comment);
> - [49] has not addressed the mosaic artifacts (called "tiling artefacts" in [49]) among individually generated patches, whereas D-VST introduces adaptive patch sampling and overlapping fusion for effective removal;
> - [49] uses multi-scale consistency objectives (spatial); D-VST employs a dual-stage learning strategy (temporal).
>
> > **C8 (Limitations)**: Maybe indeed authors can add a few sentence about the status of their model and potential limitations of using their model in clinical settings.
>
> **A8**: Thank you for your valuable suggestion.
>
> Obtaining paired cross-stain training data can be challenging and costly in real-world workflows, which may limit the scalability and applicability of D-VST.
> While such data can improve virtual staining performance with paired correspondence, unpaired data is substantially more scalable due to orders of magnitude larger amounts.
> In future work, we plan to **explore benefiting from both the scalability of unpaired data and the quality of paired data via a combination of D-VST and approaches like CycleGAN-Turbo [A], which enable diffusion models to learn from unpaired data.**
>
> [A] Parmar, Gaurav, et al. "One-step image translation with text-to-image models." (2024).

---

> > ### Comment · Reviewer_843m · 2025-08-07
> >
> > Thank you for addressing my questions, and I will keep my accept voting.

---

> > > ### Author Response · Authors · 2025-08-07
> > >
> > > ​We are pleased to know that our response has addressed your questions, and thank you very much for your thoughtful review. We truly appreciate your positive feedback.

---

### Official Review · Reviewer_3v65 · 2025-07-02

**Clarity:** 4
**Significance:** 4
**Originality:** 3
**Rating:** 6
**Confidence:** 3

**Summary:**

The authors present a diffusion-based virtual staining (VS) model specifically designed to address potential tone variation and pathology leakage in the generated images. The three main contributions of the framework are: (1) diffusion-based staining and tone control, (2) a curriculum learning method to prevent pathology leakage, and (3) a new frequency domain content-based sampling technique to produce high-quality VS images.  The analysis includes comparisons against GAN-based methods, showing superior performance compared to methods in the literature while also decreasing the computational complexity

**Questions:**

Can the authors give some timing information for large WSI generation and compare it against at least StainFuse and SynDiff

Did the authors observe any significant difference between the VS images when they used different tone domain image examples for the same sample? How robust is the model against variation of the tone domain images?

**Ethical Concerns:**

["NO or VERY MINOR ethics concerns only"]

**Final Justification:**

The authors replied to all my comments. Unfortunately, I do not see the changes in the paper since the updated version is not available in the system. If the final version of the paper is updated as described in the authors' rebuttal, I have no further reviews or comments. I will keep my score

**Limitations:**

Performance comparisons are limited to FID, PSNR, SSIM metrics, which can be diagnostically deceptive.

**Quality:**

3

**Strengths And Weaknesses:**

**Strengths**

- The authors presented a systematic approach to the problem. They tailored their solutions according to the needs of the application and drawbacks in the current methods.
- The experiments are well designed and illustrate the performance of the model well.
- Ablation studies are presented with illustrate the performance boost introduced by different components
- Patch selection strategy is well justified, provide good efficacy



**Weaknesses**

- It is not clear why the authors particularly picked SynDiff and StainFuser for comparisons. There are several other methods available in the literature (such as "Pixel super-resolved virtual staining of label-free tissue using diffusion models" or DeepLiif - rather than cycleGAN)
- Conclusions section is too short and does not contain anything more than the summary of the paper
- It would have been nice to give a WSI example in the paper, and also some timing comparison between DVST and the competing methods for virtual staining of WSI images.

---

> ### Author Rebuttal · Authors · 2025-07-31
>
> > **C1 (W1)**: It is not clear why the authors particularly picked SynDiff and StainFuser for comparisons. There are several other methods available in the literature (such as "Pixel super-resolved virtual staining of label-free tissue using diffusion models" (PSRVS) or DeepLiif - rather than cycleGAN)
>
> **A1**: Thank you for your comments.
>
> We selected SynDiff and StainFuser as they are established diffusion-based models for medical image translation and histological style transfer, respectively, aligning well with our study’s focus. They represent the diffusion-based approaches we aim to compare against.
>
> We also thank the reviewer for bringing two additional state-of-the-art approaches to virtual staining, one belonging to diffusion-based and the other to GAN-based models, to our attention.
> Following your advice, we have included PSRVS and DeepLiif (retrained per the official protocols) for comparison on the HE2IHC [51] dataset:
> | Method | DISTS↓ | FID↓ | KID↓ | PSNR↑ | SSIM↑ |
> |-------|-------|-------|-------|-------|-------|
> | D-VST (ours) | **0.154** | **33.16** | **0.0055** | **18.11±3.874** | 0.407±0.136 |
> | PSRVS | 0.422 | 230.3 | 0.2283 | 17.94±4.193 | 0.396±0.128 |
> | DeepLiiF | 0.305 | 140.3 | 0.1379 | 17.96±3.713 | **0.418±0.129** |
>
> With the due caution that these two methods may not be fully optimized for this task, we find that **our D-VST substantially outperforms PSRVS and DeepLiif in DISTS, FID, and KID,** and is comparable in PSNR and SSIM.
> We will include these results in the final version.
>
> > **C2 (W2)**: Conclusions section is too short and does not contain anything more than the summary of the paper
>
> **A2**: We appreciate the valuable feedback.
>
> In response, we will substantially revise the Conclusion section to go beyond a summary, **highlighting the broader implications, potential impact, and future research directions of our work**. We believe these additions provide a more comprehensive and meaningful conclusion to the paper.
>
> Additionally, we **supplemented the Conclusion section with a Limitations section in the Appendix**, where we
> - Discussed the multi-step denoising process inherent in diffusion models that hinders further efficiency improvement, and envisioned consistency models and distillation retraining as potential solutions;
> - Considered replacing the VAE pretrained on natural images with one fine-tuned or pretrained on histopathology images within our D-VST framework.
>
> > **C3 (W3)**: It would have been nice to give a WSI example in the paper, and also some timing comparison between DVST and the competing methods for virtual staining of WSI images.
>
> > **(Q1)**: Can the authors give some timing information for large WSI generation and compare it against at least StainFuse and SynDiff
>
> **A3**: Thank you for the constructive comments.
>
> For **WSI visualization**, we **included examples of virtually stained WSIs** (two from the HE2IHC dataset and two from the FFPE2HE dataset) in Figures A6, A7, and A8 of the Appendix.
>
> For timing, we have **recorded the generation times for the WSIs** in Figure 1 (16,384×16,384 pixels; HE to IHC) using a representative GAN-based method (pix2pixHD), two diffusion-based methods (StainFuser and SynDiff), and our D-VST.
> To avoid the unwanted mosaic artifacts, we implement the MultiDiffusion [3] sampling strategy for the compared methods, while D-VST uses its adaptive sampling strategy.
> The results are:
> | Method | Pix2pixHD | StainFuser | SynDiff | D-VST (ours) |
> |-------|-------|-------|-------|-------|
> | Time (second) | 7,127 | 840,499 | 172,032 | 8,862 |
>
> As we can see, **D-VST is orders of magnitude faster than the other diffusion-based methods and comparable to the GAN-based approach**, highlighting its efficiency advantage for large WSI virtual staining over existing diffusion-based methods.
>
> > **C4 (Q2)**: Did the authors observe any significant difference between the VS images when they used different tone domain image examples for the same sample? How robust is the model against variation of the tone domain images?
>
> **A4**: We appreciate the reviewer's insightful questions.
>
> We observed **significant differences in tones** in virtually stained images when using different tone conditioning images for the same source image.
> This is evidenced by extensive qualitative results shown in Figure 5 (main text), as well as Figures A2, A6, A7, and A8 (Appendix), where **each source image stained with a few different tone images results in apparent tone variations**, demonstrating effective tone control of D-VST.
>
> At the same time, these figures show that **the generated images consistently preserve the pathology of the source image, regardless of tone variation, thus fulfilling the pathology fidelity requirement of virtual staining**.
> For example, in Figure A6 (Appendix), images generated with tone images of different HER2 scores (with and without cancerous lesions) maintain the same HER2 score as the source, despite distinct tones.
> Thus, **D-VST is robust to tone domain variation in terms of pathology fidelity**.
>
> > **C5 (Limitations)**: Performance comparisons are limited to FID, PSNR, SSIM metrics, which can be diagnostically deceptive.
>
> **A5**: Thank you for this important point.
>
> We agree that image quality metrics (e.g., FID, SSIM) alone may not reflect diagnostic value.
> Therefore, we included a **downstream pathology classification task** (Lines 268-283), involving four HER2 scores: HER2 0: no cancerous lesions, and 1+, 2+, and 3+: increasing severity of cancerous lesions, with higher scores indicating more pronounced lesions and more advanced disease stages.
> A classifier trained on real IHC images was tested on virtual IHC images stained from HE images.
> If the virtual images are pathologically correct and realistic, classification performance should be similar to that on real ones.
>
> As shown in Table 3, **the classifier performs similarly well on both real and D-VST-generated images, and D-VST outperforms other methods by large margins**.
> This demonstrates that D-VST’s virtual images are of **high quality for potential diagnostic use**.
> Nonetheless, we acknowledge the need for more diagnosis-oriented experiments on diverse tasks to further validate clinical usability in the future.

---

> > ### Comment · Reviewer_3v65 · 2025-08-05
> >
> > I would like to thank the authors for replying to all my comments. Unfortunately, I do not see the changes in the paper since the updated version is not available in the system. If the final version of the paper is updated as described in the authors' rebuttal, I have no further reviews or comments.

---

> > > ### Author Response · Authors · 2025-08-06
> > >
> > > We sincerely thank you for your constructive feedback and for appreciating our work. We would like to clarify that the updated version of our paper is not available in the system because NeurIPS 2025 has stopped supporting the global response with PDF due to technical complications (per the official email titled "Notes about rebuttal format"). Therefore, we have focused on articulating our reply to the comments in the rebuttal. We will ensure that the final version of the paper is updated as described in the rebuttal.

---

### Official Review · Reviewer_ckuD · 2025-07-04

**Clarity:** 3
**Significance:** 3
**Originality:** 3
**Rating:** 4
**Confidence:** 3

**Summary:**

This paper presents a new method called D-VST. It is for cross-dye virtual staining of pathology images. The method uses a Diffusion Transformer. It tackles two main problems. One is "pathology leakage". This is when pathology features from a reference image wrongly appear in the result. The other problem is the "mosaic artifact". This happens when making very large Whole Slide Images (WSIs). D-VST uses two separate encoders for pathology and tone. It also uses a two-stage training plan. This helps to separate pathology from tone information. For large WSIs, the paper suggests a new patch sampling strategy. It focuses on low-frequency areas to avoid artifacts. This is also computationally efficient. The authors test their method on three tasks. The results show D-VST is better than other methods. It creates high-quality images and keeps the pathology correct. A classification task confirms this.

**Questions:**

NA

**Ethical Concerns:**

["NO or VERY MINOR ethics concerns only"]

**Final Justification:**

Thanks for the authors' detailed response. They have addressed my concerns. I have raised my score.

**Limitations:**

See the Weaknesses

**Quality:**

3

**Strengths And Weaknesses:**

**Strengths**

1.	D-VST is the first diffusion model to achieve tonal control for cross-staining virtual staining, filling a gap in existing methods.
2.	The paper uses separate encoders to extract pathology structure and staining tone. A two-stage curriculum learning first focuses on pathology, then introduces tone control to prevent “pathology leakage.” A frequency-aware adaptive patch sampling enables efficient inference on gigapixel whole-slide images and removes mosaic artifacts.
3.	The adaptive sampling method is a good innovation for large images. It solves the quality versus cost problem. This makes the tool practical.

**Weaknesses**

1.	The VAE used was pre-trained on a natural image (Stable Diffusion 1.5), which is not suitable for pathology images due to the obvious domain shift. You note in the appendix that the fine-tuning failed, but you do not analyze why, which makes the choice of VAE seem arbitrary. I suggest adding a direct performance comparison with pathology-specific VAEs (e.g., VAEs using the PathCLIP encoder) to quantify the impact of domain shifting. I would also suggest analyzing the distribution of reconstruction errors to show which specific pathology-specific structures are poorly reconstructed by current VAEs, rather than just reporting average scores.
2.	The frequency-aware adaptive sampling is a key contribution, but the choice of hyperparameters α=1 and β=8 appears inadequately justified. The paper mentions an appendix for the grid search of n, but a similar rigorous ablation study for α (which controls the sampling probability distribution ) and β (which governs the cost-quality trade-off ) is absent. The robustness and generalizability of the sampling strategy depend heavily on these parameters, and a dedicated analysis is needed to strengthen this contribution.
3.	The authors rightly note the difficulty in quantitatively evaluating billion-pixel WSIs due to the lack of suitable datasets. While understandable, this remains a limitation, especially since WSI processing is a core claim. You should suggest a way to test it. For example, you could measure consistency between patches. Or you could ask pathologists to rate the image quality blindly.
4.	The paper lacks theoretical analysis or proof and relies solely on empirical evidence. This may limit a deeper understanding of model behavior and principles, especially when explaining why two-stage learning and frequency sampling are effective.
5.	It is recommended to test the model on more diverse pathology datasets and staining types to assess its robustness and generalization capabilities. For example, images of different organs or pathology types could be included.
6.	On Ext-FFPE2HE, lower PSNR/SSIM vs. task-specific [27] attributed to "poor pixel alignment" without evidence.
7.	There is insufficient discussion of the direction of generation and the direction of medical treatment. It would have been better for the authors to present more methods [A-E].

**Minor issues**

1. Simplify long sentences for readability.
2. Check figure/table numbering and ensure all subfigures have clear captions.
3. Standardize reference formats.

**Reference**

[A] Improving generative adversarial network inversion via fine-tuning GAN encoders[J]. TPAMI, 2024.

[B] Multi-marginal wasserstein gan[J]. Advances in Neural Information Processing Systems, 2019, 32.

[C] Closed-loop matters: Dual regression networks for single image super-resolution[C]CVPR. 2020.

[D] Cross-ray neural radiance fields for novel-view synthesis from unconstrained image collections[C] ICCV 2023

[E] Segment anything model for medical image segmentation: Current applications and future directions[J]. Computers in Biology and Medicine, 2024

---

> ### Author Rebuttal · Authors · 2025-07-31
>
> > **W1**: The VAE used was pre-trained on a natural image (Stable Diffusion 1.5), which is not suitable for pathology images due to the obvious domain shift. You note in the appendix that the fine-tuning failed, but you do not analyze why, which makes the choice of VAE seem arbitrary. I suggest adding a direct performance comparison with pathology-specific VAEs (e.g., VAEs using the PathCLIP encoder) to quantify the impact of domain shifting. I would also suggest analyzing the distribution of reconstruction errors to show which specific pathology-specific structures are poorly reconstructed by current VAEs, rather than just reporting average scores.
>
> **A1**: We appreciate the reviewer's insightful comments.
>
> D-VST inherits the Stable Diffusion (SD) VAE [56] from PixArt-α [8], as it adopts the Diffusion Transformer (DiT) from PixArt-α as the denoising network.
> **Our empirical evidence, both quantitative and qualitative, shows that D-VST with the SD VAE outperforms various GAN- and diffusion-based approaches on both virtual staining and downstream classification tasks.**
> Thus, we respectfully but firmly argue that the SD VAE suffices for encoding and decoding of pathology images *in D-VST*, despite being pretrained on natural images and probably not the optimal choice.
> We attribute this to SD VAE’s strong encoding/decoding ability from large-scale, diverse training, and DiT's high compatibility with it.
> Notably, other works (e.g., HistDiST [F], StainFuser) have also successfully used VAEs pretrained on natural images for histopathology virtual staining.
>
> Meanwhile, we agree with the reviewer that further analysis is needed for insights beyond average scores.
> We fine-tuned the SD VAE on our data but observed mixed results (Appendix Lines 91-93): similar PSNR/SSIM but worse DISTS, FID, and KID.
> **Following your suggestion, we have visualized the reconstruction error distributions: the reconstructed images showed blurred cell membranes in lesion regions.**
> We will include these images and pixel-wise difference maps in the Appendix.
> We speculate this degradation is due to limited training data that prevented the VAE from learning pathological structures well, but further validation with more data is needed.
>
> Lastly, we expect that **pathology-specific VAEs could further improve D-VST's performance**.
> However, to our knowledge, no publicly available pathology-specific VAEs exist for now, and training such models (e.g., with PathCLIP encoder) is nontrivial and goes beyond the scope of this paper.
> As suggested, we plan to study the impact of domain shift via direct comparison with pathology-specific VAEs in future work.
>
> [F] Großkopf, Erik, et al. "HistDiST: Histopathological Diffusion-based Stain Transfer." arXiv preprint arXiv:2505.06793 (2025).
>
> > **W2**: The frequency-aware adaptive sampling is a key contribution, but the choice of hyperparameters α=1 and β=8 appears inadequately justified. ... The robustness and generalizability of the sampling strategy depend heavily on these parameters, and a dedicated analysis is needed to strengthen this contribution.
>
> **A2**: We agree that a dedicated analysis of α and β strengthens our contribution.
> **A rigorous ablation study with grid search was included in the Appendix (Lines 40-54, Figure A1)**, as mentioned in Lines 321-322 of the main text.
> Our analysis shows: (1) α $\in$ {0.5, 1, 2} yields similar performance, with α=1 being optimal; (2) β=8 and 16 saturate performance, and we choose 8 for lower computational cost.
> We believe this analysis justifies our choice of α and β.
>
> > **W3**: The authors rightly note the difficulty in quantitatively evaluating billion-pixel WSIs due to the lack of suitable datasets. ... You should suggest a way to test it. For example, ... Or you could ask pathologists to rate the image quality blindly.
>
> **A3**: Thank you for the constructive suggestions on evaluating WSIs.
>
> Accordingly, we have asked **two board-certified pathologists to blindly rank WSIs** virtually stained by our D-VST, a representative GAN-based method (pix2pixHD), and two diffusion-based methods (SynDiff, StainFuser), considering image quality and pathology correctness.
> The evaluated WSIs include two HE2IHC and two FFPE2HE images, shown in Appendix Figures A6, A7, and A8, respectively.
> **The mean ranking is: D-VST (1.0), pix2pixHD (2.5), StainFuser(3.0), and SynDiff (3.5)**, demonstrating D-VST’s superior virtual staining quality.
>
> > **W4**: The paper lacks theoretical analysis or proof and relies solely on empirical evidence. This may limit a deeper understanding of model behavior and principles, especially when explaining why two-stage learning and frequency sampling are effective.
>
> **A4**: We appreciate the reviewer's insightful point.
>
> As the reviewer noted in the **Strengths** comments, our main contributions are: 1) the first diffusion model for tonal control in cross-staining virtual staining, and 2) innovative adaptive sampling for efficient gigapixel WSI inference and mosaic artifact removal, which fill a gap in existing methods and make the tool practical.
> Thus, **our paper presents significant methodological contributions**.
> In addition, we provided **theoretical motivations and principles for our methodology design, supported by empirical evidence**.
>
> For the **two-stage learning** strategy (Lines 147-152): "... in Figure 2, **training with mingled pathology and tone controls often causes pathological status to leak through tone conditioning**, leading to incorrect staining outcomes. This occurs because **the model confuses the purposes of the conditioning images and mistakes the tone-conditioning image for the source of the pathology status**. Thus, **the key to effectively preventing pathology leakage is to decouple pathology and tone conditions by making the model learn both control signals precisely. For this purpose, we design a novel two-stage curriculum learning [4] scheme.**"
> In the first stage, the model learns to effectively extract and utilize the pathology information in the pathology-conditioning image.
> Then, in the second stage, the tone condition is introduced for more complex, joint learning of pathology and tone conditions.
> **This two-stage process enables effective decoupling of pathology and tone conditions**.
> Our ablation study (Table 4, w/o Curriculum) confirms that **the two-stage learning is indeed the key to pathology preservation**.
>
> For the **frequency sampling** (Lines 192–196):
> "..., we observe that **the mosaic artifact is more pronounced in low-frequency regions of the virtually stained images**. This occurs because, **unlike areas with complex textures that contain abundant clues guiding the virtual staining process, low-frequency regions require more overlapping patches for consistent denoising results**. To improve the generation quality of the low-frequency regions while simultaneously controlling computational overhead, we propose a frequency-aware adaptive patch sampling strategy (**to sample more patches in low-frequency regions**)."
> The empirical results (Appendix Lines 46-50) **validate our observation and design principle of sampling more in low-frequency regions**.
>
> > **W5**: It is recommended to test the model on more diverse pathology datasets and staining types to assess its robustness and generalization capabilities. For example, images of different organs or pathology types could be included.
>
> **A5**: We appreciate the reviewer's excellent point.
>
> **D-VST has been evaluated on datasets of two organs/cancer types**: breast cancer (HE2IHC, FFPE2HE) and colon cancer (HE2mIHC).
> Its consistent performance across cancer types demonstrates its robustness and generalization.
> We will clarify this in the final manuscript and plan to extend to more diverse tasks in the future, as the reviewer recommended.
>
> > **W6**: On Ext-FFPE2HE, lower PSNR/SSIM vs. task-specific [27] attributed to "poor pixel alignment" without evidence.
>
> **A6**: Thank you for the valuable comment.
>
> In Lines 252-253, we conjectured that D-VST's lower PSNR/SSIM versus [27] on FFPE2HE "is because the **micro-level correspondence (pixel- and structure-wise)** between the paired images in this dataset is not as good as the other two."
> This misalignment is inherent in all datasets used in our paper due to the consecutive slicing and chemical staining process, and we visually find it more serious in the FFPE2HE dataset.
> After reading this comment, we have resorted to the following procedures to **quantify the structural (mis)alignment between paired images**.
> We apply the Canny edge detector and compute the **Hausdorff distance, intersection-over-union (IoU), and Dice similarity** between edge maps of the source and target images.
> Intuitively, lower Hausdorff distance and higher IoU and Dice metrics indicate better structural alignment.
> The results on the HE2IHC and FFPE2HE datasets are:
>
> | Datasets | Hausdorff↓ | IoU↑ | Dice↑ |
> |-------|-------|-------|-------|
> | HE2IHC | **30.92±12.05** | **0.150±0.026** | **0.260±0.041** |
> | FFPE2HE | 36.63±33.13 | 0.138±0.011 | 0.242±0.016 |
>
> **FFPE2HE shows consistently worse alignment metrics, supporting our conjecture.**
> In particular, the Hausdorff distance measures the maximum deviation between two point sets, highlighting the worst-case alignment errors.
> Thus, **the substantially larger Hausdorff distances indicate more extreme misalignments.**
> Accordingly, we have also observed more poorly registered cases in FFPE2HE.
> We will add these quantitative and qualitative evidences to the Appendix.
>
> > **W7**: There is insufficient discussion of the direction of generation and the direction of medical treatment. It would have been better for the authors to present more methods [A-E].
>
> **A7**: Thank you for pointing out these excellent works.
> We will add a discussion of them to strengthen our paper.
>
> > **Minor issues**
>
> **A8**: Thank you for these suggestions. We will address all presentation issues.

---

### Decision · Program_Chairs · 2025-09-17

**Decision:**

Accept (poster)

**Comment:**

This paper presents D-VST (Diffusion Virtual Staining Transformer), a novel framework for pathology-correct, tone-controllable cross-dye virtual staining of whole slide images (WSIs). The core contributions include: 1) a dual-encoder architecture (pathology and tone encoders) combined with a two-stage curriculum learning scheme to decouple pathology and tone information, addressing "pathology leakage"; 2) a frequency-aware adaptive patch sampling strategy for efficient, high-quality inference on ultra-high-resolution WSIs, mitigating mosaic artifacts; and 3) demonstrating superior performance across three virtual staining tasks compared to state-of-the-art GAN and diffusion-based methods.

Reviewers raised concerns about VAE domain shift, hyperparameter justification, WSI evaluation metrics, theoretical analysis, and generalizability. During the rebuttal, the authors responded comprehensively and the reviewers are convinced by the authors' reponses, though theoretical gaps and paired data limitations remain to be addressed in future work.

In summary, D-VST makes impactful advancements in virtual staining, with practical implications for computational pathology. The paper is worthy of acceptance, with the expectation that the authors address the identified weaknesses in the final version (e.g., expanding theoretical motivations and discussing paired data limitations more explicitly).